# Design strategy for serine hydroxymethyltransferase probes based on retro-aldol-type reaction

Hiroshi Nonaka [1], Yuki Nakanishi[1], Satoshi Kuno[1], Tomoki Ota[1], Kentaro Mochidome[1], Yutaro Saito[1], Fuminori Sugihara[2], Yoichi Takakusagi [3,4], Ichio Aoki[3,4], Satoru Nagatoishi [5,6], Kouhei Tsumoto[5,6] & Shinsuke Sando [1,6]

Serine hydroxymethyltransferase (SHMT) is an enzyme that catalyzes the reaction that converts serine to glycine. It plays an important role in one-carbon metabolism. Recently, SHMT has been shown to be associated with various diseases. Therefore, SHMT has attracted attention as a biomarker and drug target. However, the development of molecular probes responsive to SHMT has not yet been realized. This is because SHMT catalyzes an essential yet simple reaction; thus, the substrates that can be accepted into the active site of SHMT are limited. Here, we focus on the SHMT-catalyzed retro-aldol reaction rather than the canonical serine–glycine conversion and succeed in developing fluorescent and [19]F NMR molecular probes. Taking advantage of the facile and direct detection of SHMT, the developed fluorescent probe is used in the high-throughput screening for human SHMT inhibitors, and two hit compounds are obtained.

---

[1] Department of Chemistry and Biotechnology, Graduate School of Engineering, The University of Tokyo, 7-3-1 Hongo, Bunkyo-ku, Tokyo 113-8656, Japan. [2] Core Instrumentation Facility, Immunology Frontier Research Center and Research Institute for Microbial Diseases, Osaka University, Osaka 565-0871, Japan. [3] National Institute of Radiological Sciences (NIRS), National Institutes for Quantum and Radiological Science and Technology (QST), Anagawa 4-9-1, Inage, Chiba-city 263-8555, Japan. [4] Group of Quantum-state Controlled MRI, National Institutes for Quantum and Radiological Science and Technology (QST), Anagawa 4-9-1, Inage, Chiba-city 263-8555, Japan. [5] Medical Proteomics Laboratory, Institute of Medical Science, The University of Tokyo, 4-6-1, Shiroganedai, Minato-ku, Tokyo 108-8639, Japan. [6] Department of Bioengineering, Graduate School of Engineering, The University of Tokyo, 7-3-1, Hongo, Bunkyo-ku, Tokyo 113-8656, Japan. These authors contributed equally: Yuki Nakanishi, Satoshi Kuno, Tomoki Ota. Correspondence and requests for materials should be addressed to H.N. (email: hnonaka@chembio.t.u-tokyo.ac.jp) or to S.S. (email: ssando@chembio.t.u-tokyo.ac.jp)

Folate-mediated one-carbon metabolism is a fundamental cellular process that transfers one-carbon units to multiple biochemical pathways, including the biosynthesis of purine and thymidine, the homeostasis of amino acids, such as glycine and serine, and epigenetic maintenance[1,2]. Due to its essential role in cell proliferation, the folate cycle is considered to be an effective target for drug development against rapidly proliferating cells, such as microorganisms and cancer[3,4].

Serine hydroxymethyltransferase (SHMT) has attracted attention as one of the key enzymes in folate-mediated one-carbon metabolism. SHMT catalyzes the serine–glycine conversion[1,2]. The reaction proceeds in conjunction with tetrahydrofolate (THF) and N-5,N-10-methylenetetrahydrofolate (CH2-THF) as cofactors (Fig. 1a, b). In recent years, it has been shown that SHMT expression correlates with tumor growth and prognosis[2,5,6] and the unknown physiological significance of the one-carbon unit, generated by SHMT, is associated with various diseases[7,8]. In addition, malarial SHMT has been considered to be a suitable target enzyme of parasite malaria. Therefore, SHMT has been attracting attention as a biomarker and drug target.

SHMT inhibitors have been developed with two primary biomedical goals in mind. The first goal is to develop antimalarial drugs[3,9–15]. Malaria is a life-threatening disease that spreads to people through infected anopheles mosquitoes. It has had a tremendous impact globally; 216 million people were infected in 2016, and 445,000 died[13]. In addition, the resistance of malaria parasites against existing antimalarial drugs has become a serious problem. This drug resistance problem underscores the importance of efforts to develop inhibitors for the SHMT enzyme associated with malarial parasites. The second goal is to develop anticancer drugs[1,2]. There are two main isozymes of SHMT in

mammalian cells; SHMT1 is present in the cytoplasm, and SHMT2 is present in mitochondria, as shown in Fig. 1b[4]. In chemotherapy, the three enzymes of one-carbon metabolism, SHMT, dihydrofolate reductase (DHFR)[16], and thymidylate synthase (TS)[17–19], are all potential target enzymes as this pathway is central to pyrimidine biosynthesis and is therefore strongly related to cell proliferation (Fig. 1c). In fact, inhibitors targeting DHFR and TS, such as methotrexate and fluorouracil, respectively, have been used for a long time as anticancer agents. Among the three enzymes involved in one-carbon metabolism, to our knowledge, human SHMT (hSHMT) is the only enzyme for which an established chemotherapeutic agent has not yet been developed. Therefore, hSHMT has attracted attention as a potential target enzyme for inhibitor development.

Despite the biological and medical importance, the development of molecular probes responsive to SHMT has not yet been realized. Currently, SHMT detection is carried out by a coupled enzyme assay, mainly used in a purified system, and by immuno-detection, such as blotting or ELISA using an antibody[15,20]. For example, in the case of the coupled assay for SHMT[11,15], SHMT first produces Gly and CH2-THF from Ser and THF. Then, the coupled enzyme methylene-THF dehydrogenase uses CH2-THF as a substrate to convert the coenzyme NADP+ to NADPH. By monitoring the conversion of NADP+ to NADPH by UV absorbance or fluorescence, SHMT activity can be indirectly detected.

Compared with these complicated and indirect detection methods, a facile and direct detection method for SHMT has long been desired. However, the development of molecular probes directly responsive to SHMT has been extremely difficult. This is because SHMT catalyzes a very simple reaction to convert serine

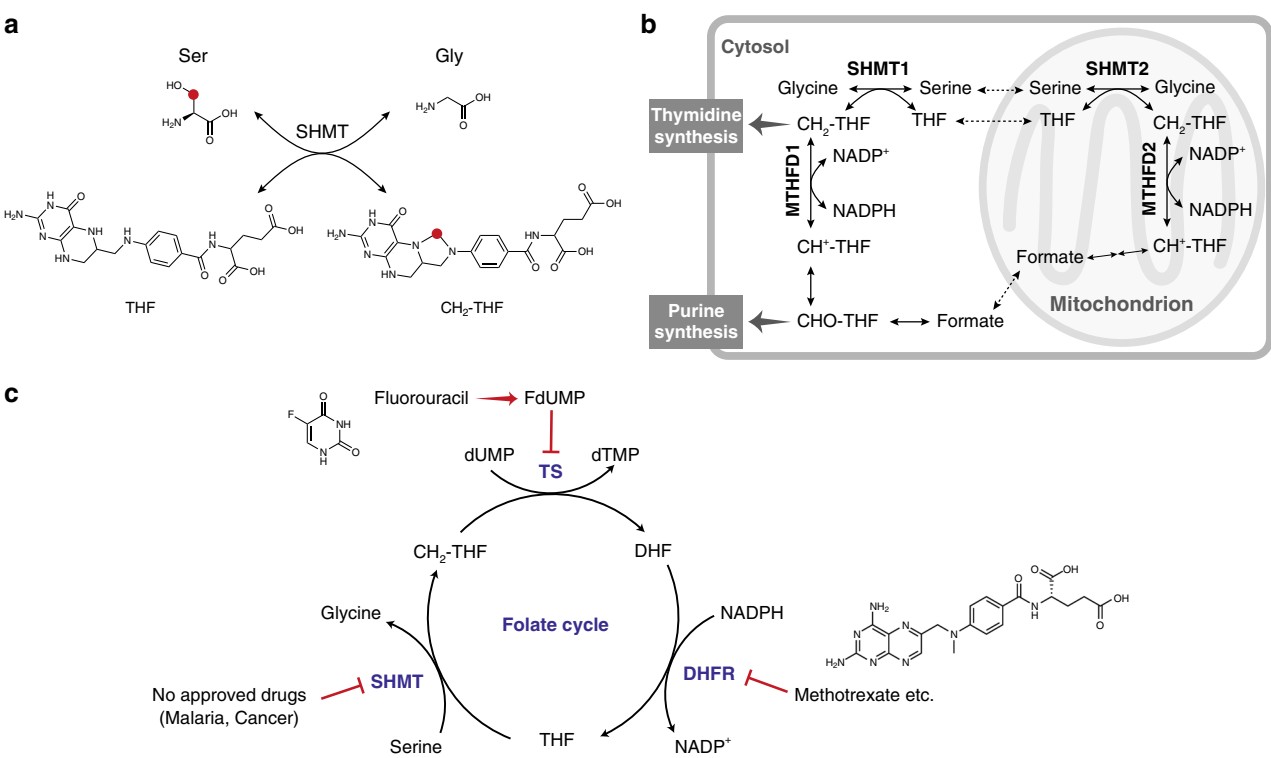

**Fig. 1** Biological role of SHMT. **a** Serine–glycine interconversion catalyzed by SHMT. THF = tetrahydrofolate, CH2-THF = N-5,N-10-methylenetetrahydrofolate. The red dot highlights the carbon that is transferred from Ser to THF. **b** Schematic overview of hSHMT function. MTHFD = methylenetetrahydrofolate dehydrogenase-cyclohydrolase, CH2-THF = N-5,N-10-methylenetetrahydrofolate, CH+-THF = 5,10-methenyltetrahydrofolate, CHO-THF = 10-formyltetrahydrofolate, NADP+ = Nicotinamide adenine dinucleotide phosphate, NADPH = NADP+ reduced form. **c** SHMT, dihydrofolate reductase (DHFR), and thymidylate synthase (TS) in the folate cycle. THF = tetrahydrofolate, CH2-THF = 5,10-methylenetetrahydrofolate, DHF = dihydrofolate, FdUMP = fluorodeoxyuridine-5′-monophosphate, dUMP = deoxyuridine monophosphate, dTMP = deoxythymidine monophosphate

to glycine and there is limited space in the substrate binding site; thus, the substrates that can be accepted into the active site of SHMT are extremely limited[21].

In this study, we undertake the challenge of developing molecular probes that target hSHMT. We focus on the retro-aldol-type reaction catalyzed by SHMT and demonstrate that various SHMT-responsive molecular probes can be designed. According to this design strategy, we achieve the development of fluorescent and [19]F NMR molecular probes for the detection of hSHMT. The developed fluorescent probe enables the sensitive and direct detection of SHMT activity, and the [19]F NMR probe is capable of detecting SHMT even under opaque biological conditions. We also use the fluorescent probe in the high-throughput screening (HTS) of hSHMT inhibitors, and two hit compounds are obtained.

## Results

**Design of hSHMT-targeting probes.** We designed molecular probes based on the enzymatic reaction mechanism and crystal structure data for SHMT. SHMT is a pyridoxal phosphate (PLP)-dependent enzyme that catalyzes the Ser–Gly conversion using the coenzymes PLP and THF (Fig. 2a, b)[21,22]. hSHMT shares 91% sequence identity with mouse SHMT and 42% sequence identity with malaria SHMT[15,21]. The active site residues of the human and mouse SHMTs are nearly identical, and the active site residues of the human and malaria SHMTs are about 80% similar[23,24]. Because human and mouse SHMTs share a high degree of homology, the structural information for mouse SHMT was used for designing probes.

Figure 2a shows the crystal structure of the (5-CHO-THF)–(Gly-PLP)–SHMT ternary complex of mouse SHMT1, which shares a high degree of homology with human and rat SHMT (Fig. 2a)[21]. Here, 5-CHO-THF acts as an analog of THF in the intermediate state. In the case of the THF-dependent pathway (R = H; upper arrow), SHMT transfers one carbon to THF from the Ser-PLP complex to afford glycine (Fig. 2b)[25]. As shown in Fig. 2a, the space surrounding the serine recognition site is very limited. This limited space at the substrate binding site hampers the development of SHMT-responsive probes.

Interestingly, it has been reported that SHMT catalyzes a THF-independent retro-aldol reaction when SHMT reacts with an aromatic β-hydroxyamino acid, such as β-phenylserine, to afford a corresponding aldehyde (R = Ar; lower arrow, Fig. 2b)[26,27]. In other words, in the THF-independent pathway, SHMT is able to accept substrates that are substituted at the β-position of serine.

With these points in mind, we conceived a design strategy for hSHMT molecular probes in which a functional reporter (R) is introduced to the β-position of serine (Fig. 2c). We decided to adopt two modalities: fluorescence and [19]F NMR. Fluorescence has the advantage of high sensitivity, and fluorescent probes have been widely used for detecting biomolecules[28–31]. NMR is also a promising detection method and has the advantage of having a high signal transparency even in opaque samples. Due to the transparency of NMR signals, NMR molecular probes are suitable for analysis under crude biological conditions. [19]F NMR, which is a nucleus with high sensitivity similar to that of [1]H, has been actively used in the design of NMR molecular probes for the analysis of biomolecules[32–40].

We designed hSHMT molecular probes in which a fluorescent or [19]F-containing aromatic moiety was introduced at the β-position of serine (Fig. 2c). The functional reporter R should fit into the binding pocket of SHMT and, most importantly, induce a fluorescent or NMR signal change upon the formation of an aromatic aldehyde, the readout for SHMT activity. We designed fluorescent probe **1** with the small compound

dimethylaminonaphthalene at the β-position of serine. It was expected to induce a change in fluorescence wavelength due to the extended conjugated system upon conversion to the aldehyde form. We also designed NMR probe **2** with 4-fluorobenzene containing a [19]F nuclei, which was expected to induce a [19]F chemical shift change due to the electronic environmental change upon conversion to the aldehyde form.

The hSHMT molecular probes **1** and **2** containing two asymmetric centers were synthesized (Fig. 3, Supplementary Figure 1, 2). Based on the tendency of substrate configurations in SHMT enzymatic reaction[15,27] and modeling, we hypothesized that the L-*erythro* form would be the optimal substrate (Supplementary Figure 3). By coupling the corresponding aromatic aldehyde and the protected glycine, by means of an aldol reaction, a fluorescent or a [19]F reporter was introduced into the β-position of serine. In the aldol reaction using lithium diisopropylamide (LDA), the *erythro* form was produced predominantly via the six-membered ring transition state. By introducing an asymmetric auxiliary group into the hydroxyl group at the β-position in the DL-*erythro*-type molecule, D and L asymmetry was separated. Based on the stereo configuration of the chiral auxiliary group, the absolute configuration of the separated probe **1** intermediate was determined by X-ray crystal structure analysis to be the L form (Fig. 3, lower inset; Supplementary Figure 4, Supplementary Table 1). Finally, the protecting groups were removed, and the desired fluorescent probe **1** and NMR probe **2** were obtained.

**hSHMT-targeting fluorescent probe.** The fluorescent probe **1** reacted with hSHMT1, and a ratiometric fluorescence intensity change was observed (Fig. 4a). When hSHMT1 was added to the solution of fluorescent probe **1**, the fluorescence intensity at 435 nm decreased and the fluorescence intensity at 530 nm increased in a time-dependent manner (Fig. 4b, excitation at 390 nm). The fluorescence values at 435 nm and 530 nm were assigned as those derived from probe **1** and dimethylaminonaphthylaldehyde (DMANA) as an expected product (Supplementary Figure 5), respectively. The product of this reaction, DMANA, was confirmed by HPLC (Supplementary Figure 6). On the other hand, when the reaction with hSHMT1 was performed in the presence of hSHMT inhibitor ((±)-SHIN1)[22], no change in the fluorescence intensity was observed. These data indicate that the fluorescence change is dependent on the hSHMT1 enzymatic reaction. In addition, the presence or absence of hSHMT1 could be detected with the unaided human eye, and hSHMT1 activity could be directly and easily detected (Fig. 4b inset).

Next, the correlation between the configurations at the two asymmetric centers and enzyme reactivity was evaluated (Fig. 4c). Upon comparing the reaction rates of hSHMT1 for the DL-*erythro* form, the DL-*threo* form, and the L-*erythro* form, it was determined that the L-*erythro* enantiomer reacted faster. These results indicate that the originally designed L-*erythro* form is the optimal substrate. The kinetic parameters of hSHMT1 for fluorescent probe **1** (L-*erythro*) were determined to be $K_m = 1.81 \pm 0.19$ mM and $k_{cat} = 0.118 \pm 0.012$ s$^{-1}$.

We then assessed potential off-target responses towards various biorelevant analytes and the cell toxicity of the aldehyde product. Almost no fluorescence change for probe **1** and the aldehyde product DMANA was observed when screened against major reactive oxygen species and reactive species containing thiols and amines (Supplementary Figure 7). To evaluate the probe stability against cytochrome P450 (CYP450) enzymes, we incubated probe **1** and DMANA with NADPH-supplemented rat liver microsomes. Under our experimental conditions, majority of probe **1** and DMANA remained intact upon incubation with the

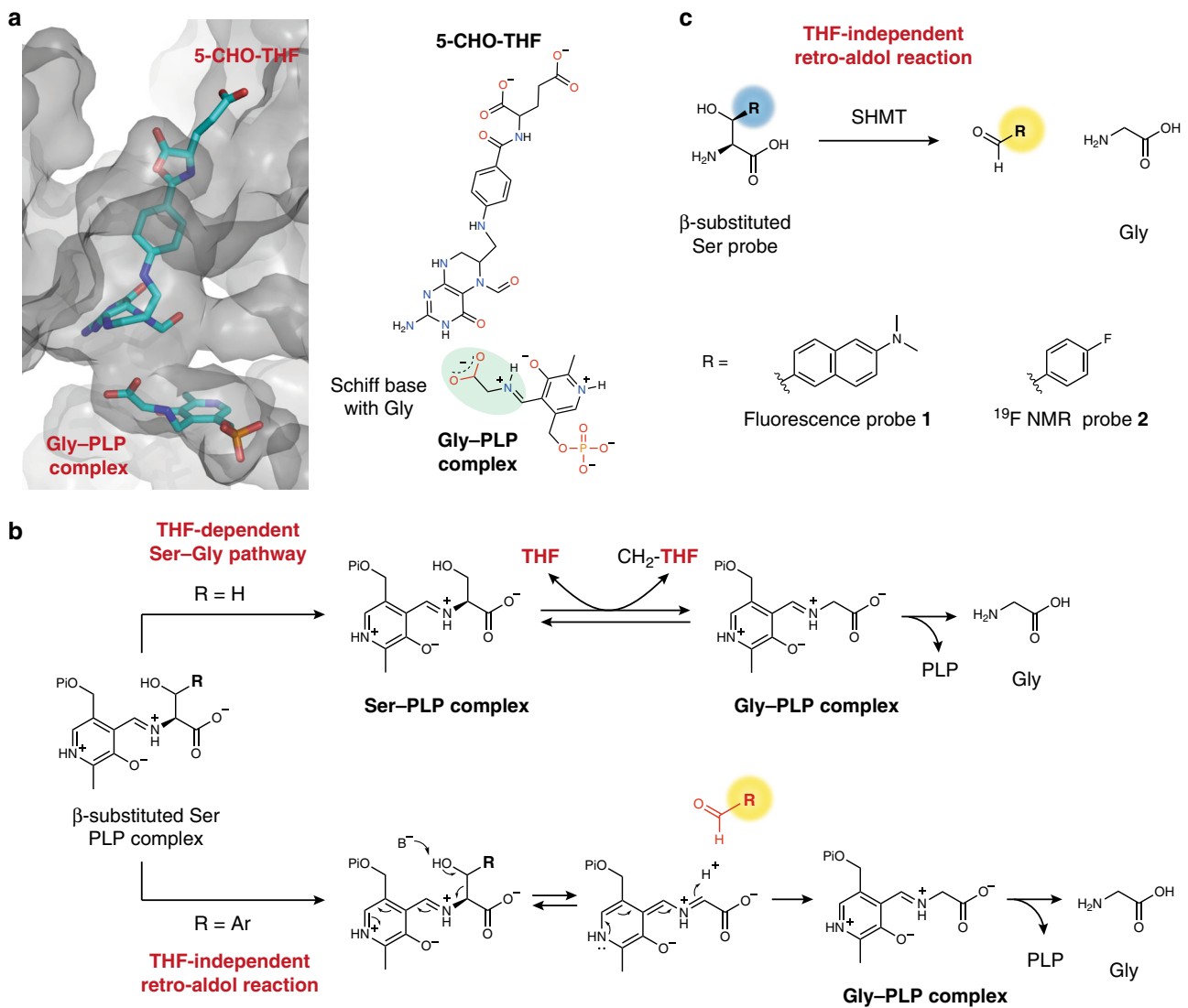

**Fig. 2** Molecular design of hSHMT probes. **a** Substrate binding site of SHMT1. Illustration of the SHMT1–5-CHO-THF–Gly-PLP complex from crystal structure data (Mouse SHMT, PDB ID: 1EJI). 5-CHO-THF and Gly-PLP are shown as stick models. Color code: oxygen: red; nitrogen: blue; carbon: cyan; phosphorus: orange. **b** The proposed mechanism of SHMT. (upper) THF-dependent serine–glycine pathway and (lower) THF-independent retro-aldol reaction catalyzed by SHMT. $Pi = -PO_3^{2-}$. **c** Fluorescent probe **1** and [19]F NMR probe **2** used in this study

microsomes (Supplementary Figure 7). We also confirmed that the aldehyde product DMANA did not exhibit cellular toxicity at low μM concentrations (Supplementary Figure 8). These experiments indicate that probe **1** is suitable for detecting hSHMT activity under biological conditions.

Finally, we checked the reactivity with isoform hSHMT2. It was found that probe **1** also reacts with hSHMT2 and exhibits a change in fluorescence (Supplementary Figure 9). Since SHMT is present in various species and has several isoforms, the pan-reactivity against SHMTs would be beneficial.

**hSHMT-targeting [19]F NMR probe**. We next evaluated the responsiveness of [19]F NMR probe **2** to hSHMT1 (Fig. 5a). Probe **2** was incubated with hSHMT1, and the mixture was subjected to [19]F NMR analysis at each time point (0–58 min). At the starting point (0 min), a single [19]F peak corresponding to probe **2** was observed at −115.3 ppm (Fig. 5b). After a 20 min incubation with hSHMT1, a signal appeared at −103.4 ppm (Fig. 5b). The latter was assigned as the 4-fluorobenzaldehyde product, after

comparison with an authentic sample (Supplementary Figure 10). After a 58 min incubation with hSHMT1, an 86% conversion of probe **2** to product was observed (Fig. 5b). Figure 5c shows the time course of conversion determined by the ratio of the [19]F NMR peak integrals. The enzymatic reaction proceeded in an incubation time-dependent manner. When the reaction with hSHMT1 was performed in the presence of the previously reported inhibitor ((±)-SHIN1, 10 μM), production of 4-fluorobenzaldehyde was suppressed completely (Fig. 5c). These data indicate that the chemical shift change of probe **2** depends on the enzymatic reaction of hSHMT1.

Using the Michaelis–Menten equation, the kinetic parameters for the enzymatic reaction of hSHMT1 with probe **2** were determined based on the change in UV absorption as $K_m = 2.75 \pm 0.27$ mM and $k_{cat} = 1.74 \pm 0.11$ s$^{-1}$. The obtained $k_{cat}$ values are higher than those of fluorescent probe **1**. Compared with the diaminonaphthyl group of fluorescent probe **1**, the 4-fluorophenyl group of [19]F NMR probe **2** was smaller, so the $k_{cat}$ values are superior. This suggests that the smaller substituent at the β-position of serine is preferable.

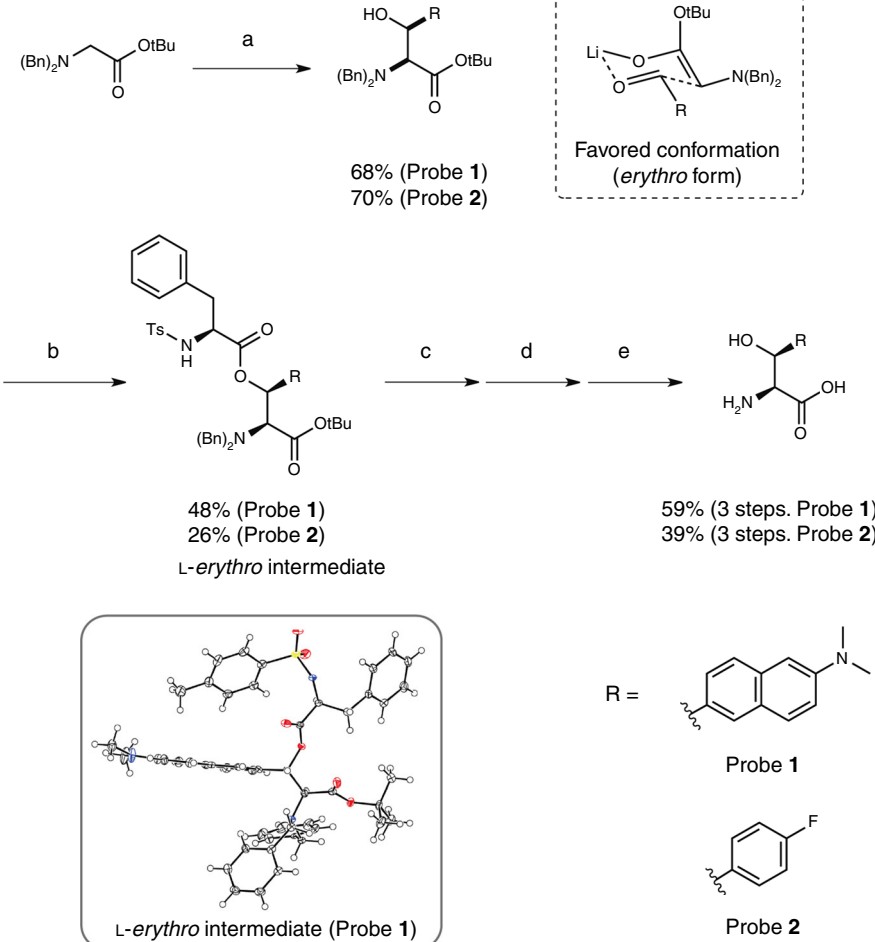

**Fig. 3** Synthesis of hSHMT fluorescent probe **1** and NMR probe **2**. Reagents and conditions for synthesis of probe **1**: a) *N*-Dibenzylglycine *tert*-butyl ester, LDA, dry THF, –60 °C, 10 min, then 6-(dimethylamino)-2-naphthaldehyde, –60 °C, 30 min; b) *N*-(*p*-Toluenesulfonyl)-ʟ-phenylalanyl chloride, DMAP, dry THF, r.t., 14 h; c) 5 M NaOH aq., THF, EtOH, r.t., 5 min; d) Pd/C, H$_2$, MeOH, r.t., 28 h; e) 4 M HCl/EtOAc, r.t. reagents and conditions for synthesis of probe **2**: a) *N*-Dibenzylglycine *tert*-butyl ester, LDA, dry THF, –78 °C, 10 min, then 4-fluorobenzaldehyde, –78 °C, 30 min; b) *N*-(*p*-Toluenesulfonyl)-ʟ-phenylalanyl chloride, pyridine, dry THF, 60 °C, 19 h; c) 5 M KOH aq., EtOH, 40 °C, 2 h; d) Pd/C, H$_2$, MeOH, r.t., 16 h; e) TFA, DCM, r.t., 17 h. The lower inset indicates the X-ray crystal structure of probe **1** ʟ-*erythro* intermediate. Color code: oxygen: red; nitrogen: blue; sulfur: yellow; carbon: black; hydrogen: white

Magnetic resonance (MR) enables the detection of signals in biological conditions such as cell lysate and tissue homogenate[40]. Taking advantage of MR analysis, we attempted to detect SHMT activity in tissue homogenate where various biological components exist. $^{19}$F NMR probe **2** was added to rat liver homogenate, which showed high SHMT expression. After a 50 min incubation with rat liver homogenate, the production of 4-fluorobenzaldehyde was confirmed by $^{19}$F NMR (50 min; Fig. 5d). When the reaction with SHMT1 was performed in the presence of SHMT inhibitor ((±)-SHIN1, 10 μM), the $^{19}$F NMR signal of the product was suppressed completely (24 h, Fig. 5d). These results indicate that the endogenous SHMT in rat liver was successfully detected. Furthermore, the $^{19}$F NMR probe **2** also worked in opaque biological samples.

With a $^{19}$F NMR probe responsive to hSHMT in hand, we turned our attention to the MR imaging of hSHMT activity. The observed chemical shift difference between $^{19}$F NMR probe **2** and the product was 11.9 ppm, which is adequate to visualize each compound selectively using $^{19}$F MR imaging. Figure 5e shows the phantom images of the hSHMT reaction sample in tubes. $^{19}$F chemical shift selective imaging (product $^{19}$F selective) produced the clear signal for the product only upon incubation with

hSHMT1 (Fig. 5e). The presence of an hSHMT inhibitor completely suppressed such a signal (Fig. 5e).

**Application to hSHMT inhibitor screening.** In recent years, inhibitors against hSHMT have attracted much attention because they could lead to the development of anticancer drugs and molecular tools for clarifying physiological phenomena related to one-carbon metabolism[1,2,4–8]. With promising molecular probes for hSHMT activity detection now in hand, HTS for hSHMT inhibitors using hSHMT molecular probes **1** and **2** was carried out[41].

We attempted to construct a fluorescent HTS system that could process a large number of samples using fluorescent probe **1**. By optimizing the assay conditions (Supplementary Table 2), a fluorescent screening system suitable for HTS, with CV$_{100\%}$ = 2.9%, CV$_{0\%}$ = 2.5%, S/B = 6.5, Z' = 0.88, was constructed. Five stages of screening using 208,233 compounds from the Drug Discovery Initiative of the University of Tokyo were conducted using the screening system (Fig. 6a). Details are described in the Supplementary Methods for high-throughput screening. Briefly, for initial screening using fluorescent probe **1**, the HTS for hSHMT inhibitors was performed and 21 candidate inhibitor

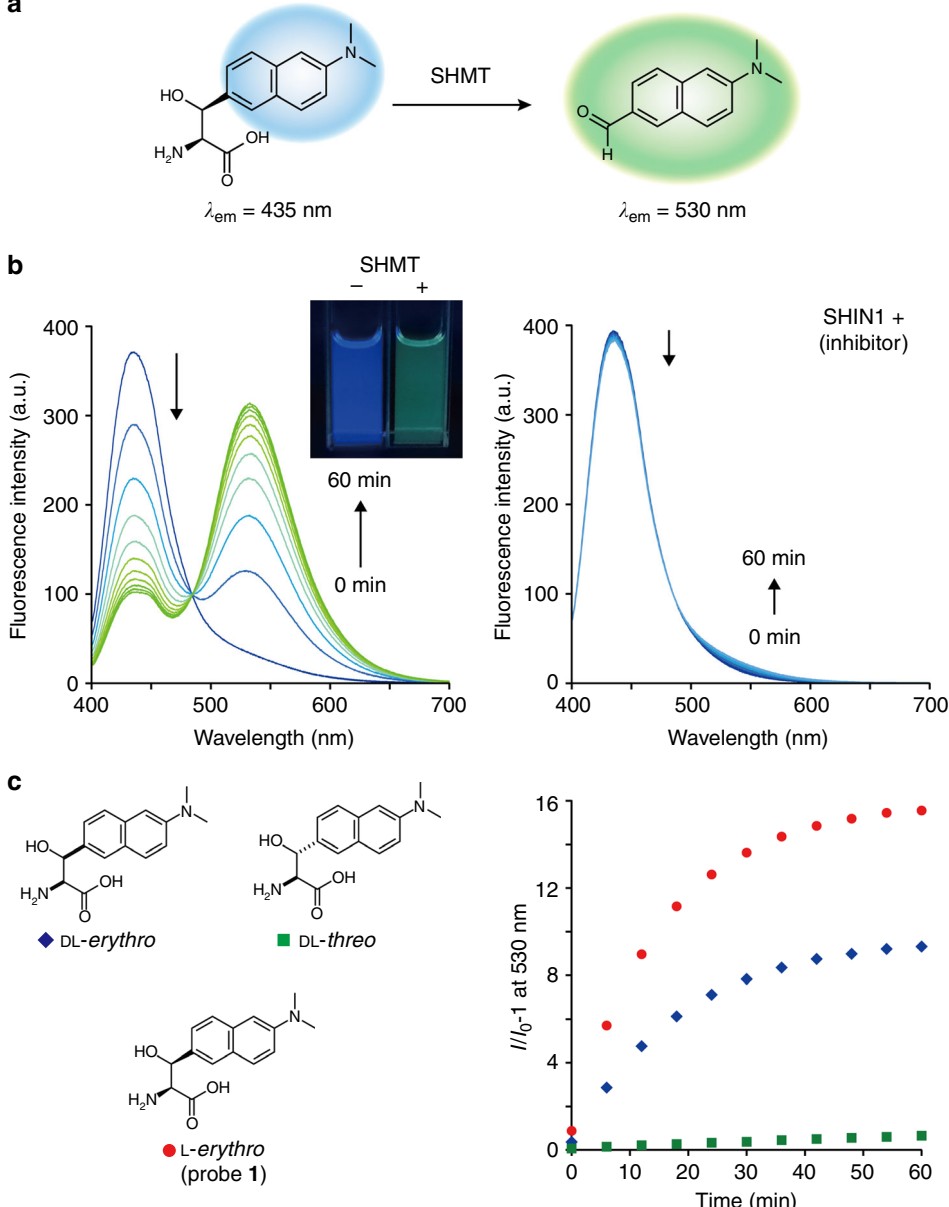

**Fig. 4** Fluorescent probe targeting hSHMT. **a** Schematic illustration of hSHMT fluorescent probe **1**. **b** Fluorescence spectral change of probe **1** (4.6 μM) during the hSHMT1-catalyzed reaction from 0 to 60 min. Excitation at 390 nm. Assay conditions: 5 units/mL hSHMT1, 50 mM HEPES buffer (pH 7.5), 100 mM NaCl, 0.5 mM EDTA, 1 mM dithiothreitol (DTT), with or without inhibitor (±)-SHIN1 10 μM, 0.6% DMSO, 37 °C. The inset shows the fluorescence change of probe **1** (5 μM). **c** Conversion rate analysis of DL-*erythro*, DL-*threo*, or L-*erythro* probes (4.6 μM) by time-dependent fluorescence analysis at 530 nm (excitation at 390 nm). $I_0$ at 530 nm is the fluorescence intensity under the condition without hSHMT1. Source data are provided as a Source Data file

compounds were detected. Subsequently, in the 2nd and 3rd screenings, the inhibitory effect of hSHMT1 on the Ser–Gly conversion reaction and selectivity via a counter assay were confirmed, respectively. Furthermore, during the 4th screening, the binding of candidate inhibitor compounds to hSHMT1 was evaluated by a thermal shift assay using differential scanning fluorimetry (DSF) (Supplementary Figure 11a). Finally, in order to confirm the efficacy of candidate inhibitors under crude biological conditions, we evaluated the inhibition of SHMT activity in mouse liver homogenate using [19]F NMR probe **2** (Supplementary Figure 12). Following the five stages of screening, two candidate compounds, Hits 1 and 2, were obtained (Fig. 6b).

The IC$_{50}$ values for Hits 1 and 2 against hSHMT1 were determined to be 0.53 μM and 0.72 μM, respectively (Fig. 6b). Both Hits 1 and 2 showed micromolar inhibiting ability against hSHMT1. Interestingly, Hits 1 and 2 possessed unique structures compared with the previously reported pyrazolopyran-based inhibitors[9–11,22,42]. These results indicate the effectiveness of the large-scale HTS system enabled by the hSHMT fluorescence probe **1**.

The specific interactions of Hits 1 and 2 with hSHMT1 were further supported by isothermal titration calorimetry (ITC). The dissociation constants were calculated to be 2.0 μM (Hit 1) and 5.7 μM (Hit 2) (Fig. 6b, Supplementary Figure 11b). Each compound showed a distinct thermodynamic profile assessed

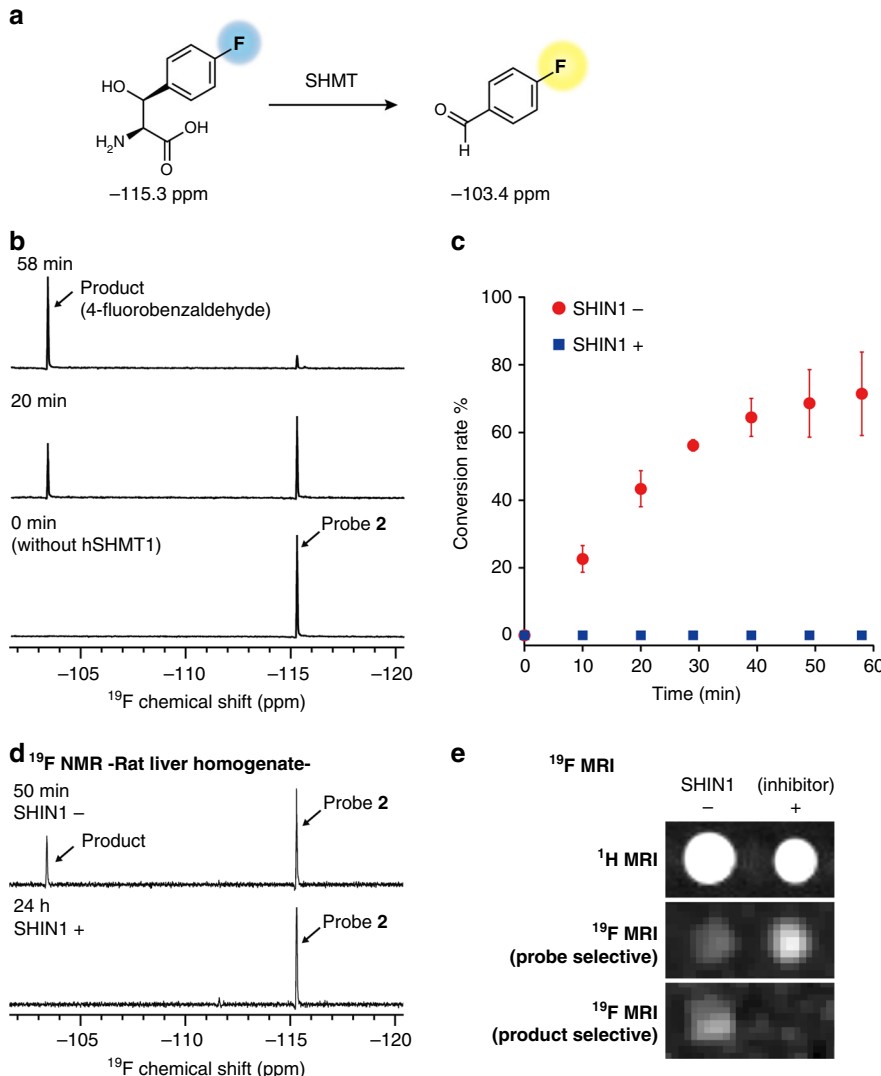

**Fig. 5** NMR probe targeting hSHMT. **a** Schematic illustration of hSHMT NMR probe **2**. **b** $^{19}$F NMR spectral change of probe **2** (5 mM) upon the addition of hSHMT1 (5 units/mL). Assay conditions: 5 units/mL hSHMT1, 50 mM HEPES buffer (pH 7.5), 100 mM NaCl, 0.5 mM EDTA, 1 mM DTT, 30% D$_2$O, with or without inhibitor (±)-SHIN1 10 µM, 0.1% DMSO, 37 °C. CF$_3$COOH (−76.5 ppm) was used as the internal standard for $^{19}$F NMR. **c** Conversion rate of hSHMT probe **2** by hSHMT1 with (blue square) or without (red circle) hSHMT inhibitor (±)-SHIN1 10 µM. Error bars represent s.d., $n = 3$. **d** $^{19}$F NMR spectra of probe **2** (1 mM) in rat liver homogenate (2.42 mg proteins/mL in PBS) with or without inhibitor (±)-SHIN1 10 µM. **e** $^1$H ($T_2$-weighted) and $^{19}$F chemical-shift-selective imaging (11.7 T) of probe **2** (10 mM). Assay conditions: hSHMT1 (1 unit/mL), with or without inhibitor (±)-SHIN1 10 µM, 37 °C, acquisition time = 17 min. Source data of Fig. 5c are provided as a Source Data file

by ITC, where the binding of Hit 1 and Hit 2 was enthalpy-driven ($\Delta H = -11.7$ kcal/mol, $-T\Delta S = 3.6$ kcal/mol, and $\Delta G = -8.1$ kcal/mol) and enthalpy/entropy-driven ($\Delta H = -2.2$ kcal/mol, $-T\Delta S = -5.2$ kcal/mol, and $\Delta G = -7.4$ kcal/mol), respectively.

In an effort to determine the inhibition mechanisms of Hits 1 and 2, the dependency of inhibition on serine concentration was evaluated. It was suggested that Hits 1 and 2 exhibit non-competitive inhibition against serine (Supplementary Figure 13).

Since there are two main isozymes of hSHMT, (hSHMT1 is present in the cytoplasm, and hSHMT2 is present in mitochondria) (Fig. 1b), the ability of Hits 1 and 2 to inhibit hSHMT2 was also evaluated. Considering the high degree of homology between hSHMT1 and 2, these compounds were expected to also act as inhibitors of hSHMT2. Indeed, both Hits 1 and 2 inhibited hSHMT2 (Supplementary Figure 14).

Finally, we attempted to inhibit the Ser–Gly conversion reaction via endogenous SHMT1 and SHMT2 in rat liver homogenate using Hits 1 and 2. As shown in Fig. 6c, Hits 1

and 2 clearly inhibited SHMT in a concentration-dependent manner. This suggests that Hits 1 and 2 may exert inhibitor activity in biological samples. The inhibitors described in this report may provide a core structure for the development of SHMT inhibitors.

## Discussion

We have succeeded in designing hSHMT molecular probes by focusing on the retro-aldol reaction activity of SHMT. The developed fluorescent and $^{19}$F NMR probes enabled the direct detection of hSHMT activity. Using fluorescent probe **1**, we conducted HTS to identify hSHMT inhibitors and succeeded in obtaining compounds possessing unique core structures.

Advantages of the strategy and the designed molecular probes are summarized below.

(1) Because of the spatial restriction of the SHMT substrate recognition site, the development of molecular probes targeting

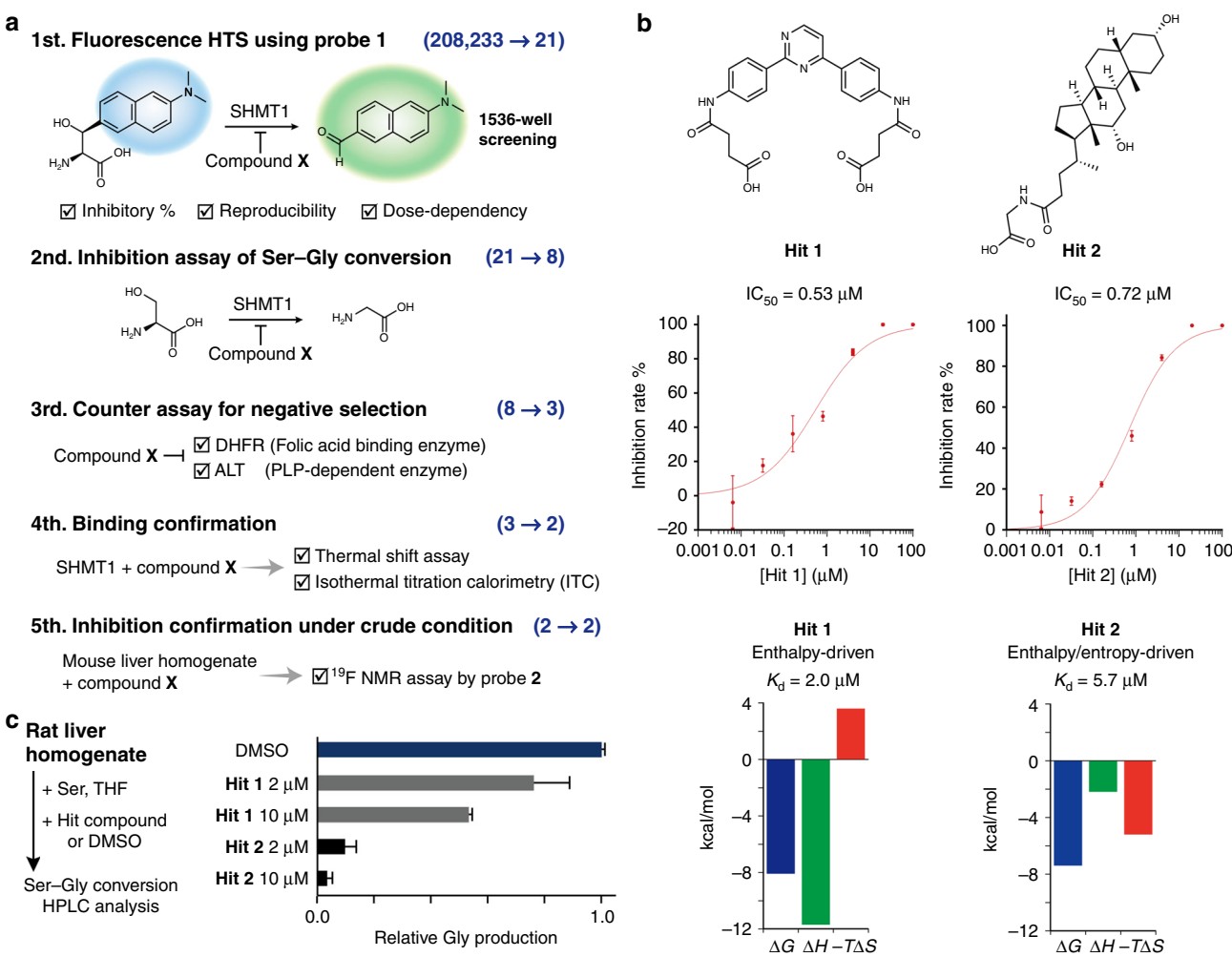

**Fig. 6** hSHMT inhibitor screening by utilizing probe **1**. **a** Schematic illustration for hSHMT1 inhibitor screening. Details are described in the Supplementary Methods for high-throughput screening. **b** Chemical structures and inhibition properties of hit compounds. The $IC_{50}$ values for Hits 1 and 2 against hSHMT1 were determined by HPLC analysis of Ser–Gly conversion in the presence of various concentrations of Hits 1 and 2. Error bars represent s.d., $n = 3$. Thermodynamic parameters of the interactions between hSHMT1 and Hit 1 or Hit 2 were determined by ITC. **c** Inhibition of Ser–Gly conversion catalyzed by endogenous SHMT by Hits 1 and 2 in rat liver homogenate (250 µg proteins/mL). Ser–Gly conversion by SHMT1 or SHMT2 in the presence of varying concentrations of Hits 1 and 2 was quantified by HPLC analysis. Error bars represent s.d., $n = 3$. Source data of Fig. 6b, c are provided as a Source Data file

SHMT has been considered a challenging goal. By focusing on THF-independent retro-aldol reaction activity rather than the canonical Ser–Gly conversion reaction, it was possible to design molecular probes that detect SHMT activity. Based on the design strategy of introducing a small functional molecule to the β-position of serine, molecular probes of various reporter methods such as fluorescence and NMR were designed without impairing hSHMT reactivity. Beyond these probes, this strategy can be used in the design of various SHMT molecular probes. In fact, using this strategy, we also developed a fluorescent turn-on probe for hSHMT, which produces a fluorescent chromophore upon reacting with hSHMT via a tandem retro-aldol–β-elimination reaction (Supplementary Figure 15, 16). This turn-on mechanism could allow researchers to design SHMT probes with various bright fluorophores, such as hydroxylcoumarin and resorufin[43,44], demonstrating the versatility of this design strategy.

(2) The substrate-based probe created based on this molecular design enabled the direct and easy detection of SHMT activity without requiring additional enzymes and reagents. The developed fluorescent and [19]F NMR molecular probes are expected to be useful molecular tools for SHMT analysis. Because additional enzymes and reagents were not required, it was possible to

analyze the inhibitor efficacy in mouse liver homogenate using [19]F NMR probe **2**, which allowed for the direct evaluation of in situ inhibitory performance in our screening system. This feature is one of the advantages of our system over a coupled assay.

(3) [19]F NMR probe **2** successfully detected endogenous SHMT in tissue homogenate and is expected to be a useful molecular tool for the detection of SHMT in biological samples. A chemical shift change of ~12 ppm is sufficient for detection with chemical shift selective [19]F MRI. By utilizing NMR, it was possible to detect SHMT in opaque samples. For in vivo applications, it will be necessary to improve the sensitivity. This may be achieved by increasing the number of [19]F atoms on the molecular probes or further enhancing the sensitivity of the MRI instrument[32,40]. Utilizing the state-of-the-art NMR-hyperpolarization technique may be another option[45,46].

(4) Because of its simplicity and facility in detecting SHMT activity, fluorescent probe **1** was used in the HTS for hSHMT inhibitors (208,233 compounds). Two hit compounds were successfully obtained. This result suggests that the large-scale compound screening system using the designed fluorescent and [19]F NMR probes is valuable. The core structures of the hit molecules

described in this report may provide a suitable foundation in the design of hSHMT inhibitors, which could be useful platforms in the design of anticancer drugs and molecular tools for clarifying physiological phenomena related to one-carbon metabolism.

## Methods

**Synthesis**. See the Supplementary Information for synthetic procedures and the characterization of compounds.

**Preparation of recombinant hSHMT**. The pET28a(+)-hSHMT1 plasmid was transformed into *E. coli* BL21(DE3)pLysS cells. Transformed *E. coli* cells were added to LB medium containing 50 μg L$^{-1}$ kanamycin and 50 μg L$^{-1}$ chloramphenicol at 37 °C. The culture was maintained overnight and diluted with 1 L of LB medium. The culture was incubated until the OD$_{600}$ reached 0.7–0.8. After cooling the medium to 25 °C, IPTG (final 0.5 mM) was added to induce expression. The culture was maintained for 20 h before harvesting by centrifugation at 3000 × *g* for 15 min at 4 °C. Cells were suspended in lysis buffer (20 mM Tris-HCl [pH 8.0], 20 mM imidazole, 300 mM NaCl, containing protease inhibitor cocktail), and then cells were disrupted by ultrasonication. The cell debris was removed by centrifugation at 12,000 × *g* for 30 min at 4 °C. The purification was conducted using Ni-NTA resin. The collected fractions were dialyzed using a 200 kDa cutoff filter (first: 20 mM Tris-HCl [pH 7.5], 300 mM NaCl, 4 °C, 1 h; second: 20 mM Tris-HCl [pH 7.5], 200 mM NaCl, 4 °C, 2 h; third: 20 mM Tris-HCl [pH 7.5], 100 mM NaCl, 4 °C, overnight). The obtained protein was analyzed by SDS-PAGE. The hSHMT1 activity was determined by the following standard protocol using DL-*erythro*-β-phenylserine as a substrate[27]. The amount of enzyme producing 1 μmol of benzaldehyde over a period of 1 h at 25 °C was defined as 1 unit. Reaction conditions: 50 mM HEPES (pH 7.5), 100 mM NaCl, 0.5 mM EDTA, 1 mM DTT, and 10 mM β-phenylserine. The recombinant hSHMT2 was also prepared following the above procedures. Before using it for experiments, the obtained hSHMT2 was incubated with PLP and dialyzed.

**Fluorescence measurements in Figs. 4b and 4c**. Fluorescence spectra were measured at 37 °C using a JASCO FP-6500 fluorescence spectrometer. The reaction buffer (5 units/mL hSHMT1, 50 mM HEPES pH 7.5, 100 mM NaCl, 0.5 mM EDTA, 1 mM DTT, 0.6% DMSO) was preincubated at 37 °C for 1 h. The fluorescent probe **1** was added to the solution and incubated at 37 °C. Fluorescence emission upon excitation at 390 nm was monitored for 60 min.

**$^{19}$F NMR measurements in Fig. 5b, c, d and e**. In Fig. 5b, c, the reaction buffer (5 units/mL hSHMT1, 50 mM HEPES pH 7.5, 100 mM NaCl, 0.5 mM EDTA, 1 mM DTT, 20% D$_2$O, 0.1% DMSO, with or without 10 μM SHIN1) was preincubated at 37 °C for 1 h. $^{19}$F NMR probe **2** was added to the solution and incubated at 37 °C. The $^{19}$F NMR spectra were measured at 37 °C (24 scans). Trifluoroacetic acid (−76.5 ppm), a counter anion of the probe, was used as the internal standard for $^{19}$F NMR. In Fig. 5d, $^{19}$F NMR probe **2** (final conc. 1 mM) was added to the rat liver homogenate (2.42 mg proteins/mL in PBS), incubated at 37 °C, and then subjected to $^{19}$F NMR analysis.

In Fig. 5e, the enzymatic reaction solution (500 μL) containing 10 mM of probe and hSHMT1 with or without 10 μM inhibitor (±)-SHIN1 was incubated at 37 °C. After 1 day, the reaction was stopped by adding 0.5 μL of 10 mM (±)-SHIN1. MR images were recorded on a BioSpec 117/11 system and processed by using ParaVision software (Bruker Biospin). The 200 μL reaction samples were dispensed into PCR tubes. The rapid acquisition with refocused echoes (RARE) method was used for $^1$H and $^{19}$F imaging. For $^{19}$F imaging, the acquisition matrix size was 64 × 32 (zero filled to 128 × 64 for image processing) with a field of view of 8.0 × 4.0 cm. The repetition time was 1000 ms, and the effective echo time was 48.0 ms. The number of accumulations was 128.

**Animal-derived materials**. Mice and rats were maintained in accordance with the guidelines of the National Institute of Radiological Sciences (NIRS), and all experiments were reviewed and approved by the NIRS committee for the care and use of laboratory animals.

**Reporting summary**. Further information on experimental design is available in the Nature Research Reporting Summary linked to this article.

## Data availability
The source data underlying Figs. 4b–c, 5c and 6b–c, sequence data of expressed SHMT, and Supplementary Figs 7–9 and 13–15 are provided as a Source Data file. The X-ray crystallographic data reported in this study have been deposited at the Cambridge Crystallographic Data Centre (CCDC), under deposition number CCDC 1873543. This data can be obtained free of charge from The Cambridge Crystallographic Data Centre via www.ccdc.cam.ac.uk/data_request/cif. The data that support the findings of this study are available from the corresponding authors upon reasonable request.

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

## Acknowledgements

We thank Dr. Takayoshi Okabe of The University of Tokyo for discussions and assistance with inhibitor screening, and Dr. Shuhei Kusumoto for his assistance with X-ray crystal structure analysis. This work was supported by CREST (JPMJCR13L4), Japan Science and Technology Agency (JST). This work was partially supported by Platform Project for Supporting Drug Discovery and Life Science Research (Basis for Supporting Innovative Drug Discovery and Life Science Research (BINDS)) from Japan Agency for Medical Research and Development (AMED) under Grant Number JP18am0101094. This work was partially supported by the Platform Project for Supporting Drug Discovery and Life Science Research from AMED under Grant Number JP17am0101086. Plasmid for recombinant hSHMT2 including A269T mutation was a gift from Prof. C. Arrowsmith of University of Toronto, Canada (Addgene plasmid # 25479). Plasmid for recombinant eDHFR was a gift from Prof. S. Tsukiji of Nagoya Institute of Technology, Japan.

## Author contributions

H.N. and S.S. initiated and designed the project. Y.N., T.O., K.M. and Y.S. performed the probe syntheses. H.N., Y.N., S.K., T.O., K.M., Y.S., F.S., Y.T. and I.A. performed fluorescence and NMR/MRI experiments. Y.N. and S.K. conducted the HTS experiment. S.K., T.O., K.M, S.N. and K.T. conducted DSF and ITC of the hit compounds. The manuscript was written by H.N. and S.S. and edited by all the coauthors.

## Additional information

**Competing interests:** The authors declare no competing interests.

