## [Peer Review File · Nature Communications]

Reviewers' comments:

Reviewer #2 (Remarks to the Author):

The manuscript under consideration by Sando and co-workers describes the development of fluorescent and ^{19}F probes for serine hydroxymethyltransferase (SHMT) activity. The general design strategy exploits the retro-aldol-type reaction to generate an aldehyde product. In the case of the fluorescent probe the authors selected a N,N-dialkyl naphthalene dye which upon reaction with SHMT generates an aldehyde that is in conjugation with the electron donating moiety. This leads to a change in the fluorescent properties of the probe. Likewise, the ^{19}F probe relies shifting the fluorine resonance upon reaction, a strategy introduced by Ralph Mason and co-workers over a decade ago (10.1016/j.mri.2006.04.003). The authors claim their probes hold distinct advantages over existing methods such as coupled enzyme assays. The authors also noted that their probes contain two asymmetric centers and thus motivated them to synthesize the various isomers to elucidate which form would be the optimal substrate for SHMT. The author complimented these studies by determining the rates for each isomer. The author then performed NMR / MRI studies using the ^{19}F probe and showed a difference in signal when comparing the unactivated probe and the SHMT-treated probe. Finally, the authors employed their fluorescent probe in a screen to identify 2 inhibitors of SHMT activity. Overall this study was nice but as a reviewer I do not believe it is at the caliber of Nature Communications. The authors are urged to consider resubmitting to a different journal after addressing the points below.

1) One of the major advantages of having a direct SHMT probe over existing methods such as the coupled-enzyme assay is that it can be used on intact samples (e.g., live cells). However, in each of the instances the authors selected to use the probe in vitro or in homogenates. I suspect the issue is that the probes are not cell permeable and thus, the authors were constrained to working around this limitation. If my suspicions are correct the authors should mask the carboxylate with a chemical group such as AM ester which can then be removed by intracellular esterases.

2) The authors mentioned there are two SHMT isoforms, why was this only tested against only one of the isoforms? How can you ensure that you are not getting pan-reactivity? And if you do have reactivity with both (which should be the case based on the proposed mechanism) this decreases the utility of the probe because in each instance you will need to do much more elaborate control experiments with siRNA KO etc.

3) The authors were not careful in characterizing the stability of the probe or of the turned over product. This was a major sticking point for me. In particular, are there enzymes found in living systems can activate or change the properties of the probe? The authors should at least test this against CYPs. More concerning, however, is the stability of the turned over aldehyde products. The authors made no attempts to evaluate its stability after the product has been formed. Fluorogenic aldehyde probes can form Schiff-bases, iminiums, they can cyclize with Cys and homo-Cys (see aldehyde based probes for these amino acids) and they can be oxidized to the acid product by a variety of ROSs. Since the authors claim their probe will likely find utility in cancer, oxidative stress in tumors is a major problem.

4) When I review papers for high profile journals such as Nat Commun, I consider whether the paper can spark new discovery in various fronts. From a probe perspective the current design is limited to only producing aldehyde products. The major problem with this is that aromatic aldehydes are typically non-fluorescent due to donor-PeT quenching. This means any other dye scaffold other than the naphthalene used by the authors would result in an on-off response. Turn-off probes are not desirable in the community because other factors can lead to a decrease in signal such as dye efflux, bleaching etc. In fact, this is likely why the authors decided to make a ^{19}F probe instead of other fluorescent analogs because the MRI version is not impacted by quenching. I thought this was clever and resourceful.

5) While it may seem on first glance that the authors were being scientifically rigorous by determine which isomer is the best substrate. These experiments are simply not important, nor do add significantly to the paper. If this information was responsible for helping the authors design the current probe that is a different story. As is, it is just distracting. It is similar to how a lot of probe-based papers add DFT calculations that have no true value beyond bulking up the study.

6) Developing ^{19}F MRI probes is an area that our group has strongly considered in the past. However, there were several important reasons that ultimately caused us to not pursue this area. Firstly, ^{19}F is not used clinically and it does not seem like this will happen in the future. The issue is the poor sensitivity. If the authors compare the ^1H MRI images with their ^{19}F images it is abundantly clear this is the case. Most MRI facilities do not have coils that can detect ^{19}F which means this is really a niche area. Lastly, having talked to a lot of MRI experts in the field they all told us that dynamic probes are 'neat and cool' but not desirable for real translational applications. Since you already have an issue seeing any signal at all, imagine if 20% of that signal is shifted to another species. How are you going to see that?

7) I understand it was necessary to do something more with the probe to try make the study more comprehensive. I would have preferred the cell studies mentioned above but because it does not seem like the probes are compatible with live systems, the authors decided to perform a screen. You can do exactly the same screen using the coupled enzyme assay since it is *in vitro*. While the coupled assay may be operationally more complex, it can still be done. This brings us back to the question of whether the current probes have made a big enough of an advance to warrant publication in a high profile journal.

Reviewer #3 (Remarks to the Author):

Review

Recommendation: Publish in Nature Comm. after minor revisions.

Comments:

The manuscript „Design Strategy for Serine Hydroxymethyltransferase Probes Based on Retro-Aldol-Type Reaction“ from the Prof. Sando et al. describes the development of chemical probes for the enzyme SHMT as markers for the application in fluorescence and ^{19}F NMR spectroscopy. Furthermore, they were able to demonstrate the convenience and benefit of their methods by e.g. applying the chemical probes to a high-throughput screening to identify two potential lead structures for drug development, which is a highlight of the paper.

This work nicely illustrates the use of a SHMT-induced retro-aldol reaction to access either fluorescent or ^{19}F labelled chemical probes on a rather complex target. Especially, the ^{19}F labelled chemical probe was shown to be a potential tool for the SHMT detection in biological samples. Overall the work is of high quality and merits the publication in Nature Comm. It will find a broad audience that includes pharmacologists, medicinal chemists, and biochemists. This publication will fill a gap in the development of SHMT-related potential drugs and will contribute to a success in this field.

Major revisions:

Despite the great science in this work, the major weakness of this manuscript is the writing, which does not appear to meet the standard of Nature Comm. and needs some improvement. Furthermore, the literature is not appropriately covered in the introduction part, which is already proven by “only” 19 citations in total.

1) The fields of fluorine MRS and MRI in biomedicine and fluorescence spectroscopy are inadequately represented missing impactful but also historically important reviews and articles (e.g. 1. Ahrens et al. "In vivo imaging platform for tracking immunotherapeutic cells." Nature Biotechnology 23: 983. 2. Ruiz-Cabello J, Barnett BP, Bottomley PA, Bulte JWM. Fluorine ^{19}F MRS and MRI in biomedicine. NMR in Biomed. 2011;24:114-29. 3. Ahrens ET1, Bulte JW. Tracking immune cells in vivo using magnetic resonance imaging. Nat Rev Immunol. 2013;13:755-63. 4. Weiss, S. "Fluorescence Spectroscopy of Single Biomolecules." Science 1999, 283 (5408): 1676-1683...). Those are just a couple examples but there are many more. Therefore, the introduction requires minor corrections and highlighting of the chosen methods and their advantages.

2) The research on the target SHMT does not only include the therapeutic area of cancer but also

Malaria, which needs to be mentioned and introduced, because the pyrazolopyran-based inhibitors reported in the manuscript were also used against *Plasmodium falciparum* (e.g. 1. Maenpuen, S.; Sopitthummakhun, K.; Yuthavong, Y.; Chaiyen, P.; Leartsakulpanich, U. Characterization of *Plasmodium falciparum* serine hydroxymethyltransferase—A potential antimalarial target *Mol. Biochem. Parasitol.* 2009, 168, 63– 73; 2. Sopitthummakhun, K.; Maenpuen, S.; Yuthavong, Y.; Leartsakulpanich, U.; Chaiyen, P. Serine hydroxymethyltransferase from *Plasmodium vivax* serine is different in substrate specificity from its homologues *FEBS J.* 2009, 276, 4023– 4036....). The introduction might cover more information about the target SHMT and the approaches to address this target.

3) The entire manuscript requires some harmonization and standardization. (examples are given below).

4) The authors could comment on possible side-effects and toxicity of the corresponding aldehydes especially Schiff-base formation *in vivo*.

5) Please further elucidate the homology especially of the active site between the three species. How adaptable is the mouse model to humans?

6) Please comment on the feasibility of the compounds for further use in *in vivo* experiments. Is imaging in mice/rats possible?

7) The modelling process needs some explanation. Was there a program used or were the ligands manually placed? Otherwise this computer-supported experiment needs to be made to prove the hypothesis of the size-dependent kinetic results.

Minor revisions:

Page 4, Caption 1: ...SHMT. a) Serine-glycine ... THF. b) Schematic ...

Page 4, Caption 1: Red circle highlights the carbon ...

Page 4, Caption 1: two abbreviations for 5,10-methylene-tetrahydrofolate (CH₂-THF and 5,10-CH₂-THF), please keep consistency.

Page 4, Caption 1: upper case for positive charge of NADP⁺.

Page 6, line 6: Figure 2a shows ...

Page 6, line 10: Figure 2b shows ... (redundant to the previous section and bumpy to read).

Page 6, line 10: The sentence "When SHMT ...to afford glycine" requires rewriting.

Page 6, line 13: "crowded" colloquial writing

Page 6, line 17: "When considering..." on the one hand redundant and on the other stylistically poor written

Page 6, line 18: The sentences on this page require rewriting.

Page 7, Figure 2: ...probes. a Substrate... ...b ... c... (these minor errors needs to be corrected in the entire manuscript).

Page 7, Figure 2: PDB ID: 1EJI in capital letters

Page 7, Figure 2 : Oxygen: red ; nitrogen: blue ...

Page 9, Figure 3 ; From a chemist point of view, the scheme should contain the temperatures, rxn times, and the deprotection conditions in detail, or should at least be mentioned in the caption of the Figure.

Page 9, Figure 3: Oxygen: red; nitrogen: blue...

Page 10, line 15: "naked eye" colloquial writing

Page 10, line 17-21: redundant and very difficult to read.

Page 10, line 22: "The kinetic parameters ..." sentence needs to be rewritten

Page 11, Figure 4: see comments above: a... b...

Page 14, Figure 5: Misspelling of "inhibitor" in e) (see comments above)

Page 15, line 17: ...selectivity by a counter assay

Page 15, line 21: ...Hit 1 and 2...

Page 16, line 23: ...both of the Hit compounds...

Page 17, Figure 6: b) (Top) ... and (bottom) ...

Page 19, line 2: We have succeeded, for the first time, the design of SHMT molecular probes by focusing...

Page 19, line 11: "...SHMT has been thought to be difficult." Sentence sounds colloquial, a better sentence might be: ...considered as a challenging goal.

Response to the referees' comments and revisions that have been made

We thank all of the reviewers for their comments. These have been very helpful in further improving the manuscript. We revised our manuscript in lights of all the comments as follows.

Reviewer #1

Original comments from Reviewer #1

In this submission to *Nature Communications*, Sando and coworkers describe probes that allow for convenient fluorescence-based and ¹⁹F NMR-based direct readout assays for SHMT (serine hydroxymethyltransferase) activity. This is cleverly done; the authors have taken advantage of the ability of this enzyme to catalyze a retro-aldol reaction with erythro-beta-aryl substituted L-serine analogues independent of the tetrahydrofolate (THF) cofactor. This allows for direct release of either a fluorescent or a ¹⁹F-containing aromatic aldehyde, and its immediate detection by a plate-reading fluorimeter or by NMR, respectively. Such assays are useful chemical biological tools for the rapid screening of the effects of modulators or PTM (post-translational modification) on the activity of native SHMT, and, as such, will certainly draw the attention of many in the chemical biology community. The authors also demonstrate that this new assay serves as a useful platform for parallel screening in the search for new SHMT inhibitors.

Moreover, they demonstrate proof of principle here by utilizing the new platform to identify two new hit scaffolds with submicromolar IC₅₀s. The authors use complementary assays (ITC, thermal denaturation, etc) to verify these hits, and they also demonstrate that both compounds inhibit the physiologically important L-serine to glycine plus N⁵,N¹⁰-CH₂-THF reaction. Temperature dependence of inhibitor binding is also studied, allowing the authors to sort out entropic vs. enthalpic contributions.

Because the literature indicates that the L-*erythro*-stereoisomers are the best retro-aldolase substrates for SHMT, the authors are careful to target these isomers. Relative stereochemistry is controlled by utilizing an N,N-dibenzylglycinate enolate that tends to give the E-enolate geometry, thereby providing predominantly the *erythro* (*anti*) aldol product. Subsequent chiral derivatization allows the authors to generate separable diastereomers and thereby gain access to desired L-*erythro*-stereoisomer as well as establish the absolute stereochemistry via extra crystallography.

All in all, this is a very nice study, and, in my view, with a few modifications and enhancements of the discussion, such a piece would appeal to the wide-ranging readership of

Nature Communications.

The authors should discuss clearly that the biologically relevant L-Ser to Gly + N5,N10-CH₂- THF reaction can be assayed through a coupled reaction with methylene-THF dehydrogenase, as was done in the studies by Diederich and Chaiyen, for example (*J. Med. Chem.* – **2017** - Reference 17 and *ChemMedChem* **2018**, *13*, 931-934 (please add this reference!). This results in the conversion of NADP⁺ to NADPH which can be observed by UV absorbance at 340 nm or, in principle, by fluorescence emission at ~460 nm. This would be a two-enzyme coupled alternative to the assay presented in this paper.

The authors should also highlight the importance of developing inhibitors of the SHMT enzyme more clearly and comprehensively. Beyond being a validated target for malaria (*Plasmodium falciparum* enzyme), to my knowledge, the human enzyme remains the only enzyme of the three enzyme one carbon cycle for which established chemotherapeutic agents have not yet been developed. The authors should discuss the importance of inhibitors of the other two enzymes of the one carbon cycle, DHFR (dihydrofolate reductase) and TS (thymidylate synthase), for chemotherapy. As for the 19-F assay, the authors should add a broader discussion of previous uses of 19-F NMR to study PLP enzyme inhibition. Some references for these discussion points are suggested below:

Recent reference on dihydrofolate reductase DHFR inhibition for anti-cancer therapeutics development:

Ng, Hui-Li; Ma, Xiang; Chew, Eng-Hui; Chui, Wai-Keung “Design, synthesis, and biological evaluation of coupled bioactive scaffolds as potential anticancer agents for dual targeting of dihydrofolate reductase and thioredoxin reductase,” *J. Med. Chem.* **2017**, *60*, 1734-1745.

Recent references on thymidylate synthase (TS) inhibition for anti-cancer therapeutics development:

Negrei, Carolina; Hudita, Ariana; Ginghina, Octav; Galateanu, Bianca; Voicu, Sorina Nicoleta; Stan, Miriana; Costache, Marieta; Fenga, Concettina; Drakoulis, Nikolaos; Tsatsakis, Aristidis M. “Colon cancer cells gene expression signature as response to 5-

fluorouracil, oxaliplatin, and folinic acid treatment,”

Frontiers in Pharmacology **2016**, *7*, 172/1-172/10

Azijli, Kaamar; van Roosmalen, Ingrid A. M.; Smit, Jorn; Pillai, Saravanan; Fukushima, Masakazu; de Jong, Steven; Peters, Godefridus J.; Bijnsdorp, Irene V.; Kruyt, Frank A. E. “The novel thymidylate synthase inhibitor trifluorothymidine (TFT) and TRAIL synergistically eradicate non-small cell lung cancer cells.”

Cancer Chemotherapy and Pharmacology **2014**, *73*, 1273-1283.

Gomez-Martin, C.; Salazar, R.; Montagut, C.; Gil-Martin, M.; Nunez, J. A.; Puig, M.; Lin, X.; Khosravan, R.; Tursi, J. M.; Lechuga, M. J.; et al. “A phase I, dose-finding study of sunitinib combined with cisplatin and 5-fluorouracil in patients with advanced gastric cancer,”

Investigational New Drugs **2013**, *31*, 390-398.

Discussions of the use of fluorinated probes for PLP enzyme activity:

Berkowitz, David B.; Karukurichi, Kannan R.; Salud-Bea, Roberto; Nelson, David L.; McCune, Christopher D. “Use of Fluorinated Functionality in Enzyme Inhibitor Development – Mechanistic and Analytical Advantages” *J. Fluor. Chem.* **2008**, *129*, 731-742.

Karukurichi, Kannan R.; Salud-Bea, Roberto; Jahng, Wan Jin; Berkowitz, David B. “Examination of the New alpha-(2’Z-Fluoro)vinyl Trigger with Lysine Decarboxylase: The Absolute Stereochemistry Determines the Reaction Course,” *J. Am. Chem. Soc.* **2007**, *129*, 258-259.

Point-by-point response to the comments of Reviewer #1

We wish to thank the reviewer for the comments and for providing constructive suggestions. We have addressed all the points raised by the reviewer through new text.

- ✓ **Comment.** The authors should discuss clearly that the biologically relevant L-Ser to Gly + N⁵,N¹⁰-CH₂- THF reaction can be assayed through a coupled reaction with methylene-THF dehydrogenase, as was done in the studies by Diederich and Chaiyen, for example (J. Med. Chem. – 2017 - Reference 17 and ChemMedChem 2018, 13, 931-934 (please add this reference!). This results in the conversion of NADP⁺ to NADPH which can be observed by UV absorbance at 340 nm or, in principle, by fluorescence emission at ~460 nm. This would be a two-enzyme coupled alternative to the assay presented in this paper.

>>>Response to Comment.

According to the reviewer's comment, we have added new sentences and references about an assay system of SHMT enzymatic reaction utilizing a coupled reaction with methylene-THF dehydrogenase.

[Revised manuscript, p.7, lines 4–9]

For example, in the case of SHMT coupled assay,^{11,15} SHMT firstly produces Gly and CH₂-THF from Ser and THF. Then, the conversion of coenzyme NADP⁺ to NADPH is caused by a coupled enzyme methylene-THF dehydrogenase using the CH₂-THF as a substrate. By monitoring this NADP⁺–NADPH conversion with UV or fluorescence, SHMT activity can be indirectly detected.

- ✓ **Comment.** The authors should also highlight the importance of developing inhibitors of the SHMT enzyme more clearly and comprehensively. Beyond being a validated target for malaria (*Plasmodium falciparum* enzyme), to my knowledge, the human enzyme remains the only enzyme of the three enzyme one carbon cycle for which established chemotherapeutic agents have not yet been developed.

>>>Response to Comment.

We thank the reviewer for the constructive suggestion. Following the reviewer's comment, we have changed the main text so that the significance of development of SHMT inhibitors is clearly and comprehensively explained.

[Revised manuscript, p.4, line 13 – p.5, line 3]

The development of SHMT inhibitors has been performed especially toward treatment of two types of diseases. The first one is antimalarial drug.⁸⁻¹⁵ Malaria is a life-threatening disease that spreads to people through infected anopheles mosquitoes. It has a tremendous impact globally, 216 million people are infected in 2016, and 445,000 people have died.¹³ In addition, the resistance of malaria parasites against existing antimalarial drugs has become a serious problem. Under such circumstances, researches and investigations for new inhibitors against malaria SHMT has been conducted. The second one is anticancer drug.^{1,2} In chemotherapy, three enzymes of one-carbon metabolism, SHMT, dihydrofolate reductase (DHFR),¹⁶ and thymidylate synthase (TS)¹⁷⁻¹⁹ are potent target enzymes strongly related to cell proliferation (**Fig. 1c**). Actually, inhibitors targeting DHFR and TS, such as methotrexate and fluorouracil respectively, have been used for a long time as anticancer agents. Among the three enzymes of the one-carbon metabolism, to our knowledge, human SHMT is the only enzyme to which an established chemotherapeutic agent has not yet been developed. Therefore, human SHMT has attracted attention as a target enzyme of anticancer drug.

- ✓ **Comment.** The authors should discuss the importance of inhibitors of the other two enzymes of the one carbon cycle, DHFR (dihydrofolate reductase) and TS (thymidylate synthase), for chemotherapy.

>>>Response to Comment.

We agree with the reviewer's constructive comment, we have added a new sentence and figure (**Fig. 1c**) showing the importance of inhibitors of DHFR and TS in chemotherapy.

[Revised manuscript, p.4, line 13 – p.5, line 3]

The development of SHMT inhibitors has been performed especially toward treatment of two types of diseases. The first one is antimalarial drug.⁸⁻¹⁵ Malaria is a life-threatening disease that spreads to people through infected anopheles mosquitoes. It has a tremendous impact globally, 216 million people are infected in 2016, and 445,000 people have died.¹³ In addition, the resistance of malaria parasites against existing antimalarial drugs has become a serious problem. Under such circumstances, researches and investigations for new inhibitors against malaria SHMT has been conducted. The second one is anticancer drug.^{1,2} In chemotherapy, three enzymes of one-carbon metabolism, SHMT, dihydrofolate reductase (DHFR),¹⁶ and thymidylate synthase (TS)¹⁷⁻¹⁹ are potent target enzymes strongly related to cell proliferation (**Fig. 1c**). Actually, inhibitors targeting DHFR and TS, such as methotrexate and fluorouracil respectively, have been used for a long time as anticancer agents. Among the three enzymes of the one-carbon metabolism, to our knowledge, human SHMT is the only enzyme to which an established chemotherapeutic agent has not yet been developed. Therefore, human SHMT has attracted attention as a target enzyme of anticancer drug.

Figure 1 | Biological role of SHMT. **a)** Serine–glycine conversion catalyzed by SHMT. THF = tetrahydrofolate, CH₂-THF = 5,10-methylene-tetrahydrofolate. Red circle highlights one carbon that is transferred from Ser to THF. **b)** Schematic overview of SHMT function. MTHFD = methylenetetrahydrofolate dehydrogenase-cyclohydrolase. CH₂-THF = 5,10-methylene-tetrahydrofolate. CH⁺-THF = 5,10-methenyltetrahydrofolate. CHO-THF = 10-formyl-tetrahydrofolate. NADP⁺ = Adenine-nicotinamide dinucleotide phosphate. NADPH = NADP⁺ reduced form. **c)** SHMT, dihydrofolate reductase (DHFR), and thymidylate synthase (TS) in folate cycle. THF = tetrahydrofolate, CH₂-THF = 5,10-methylene-tetrahydrofolate, DHF = dihydrofolate, FdUMP = fluorodeoxyuridine-5'-monophosphate, dUMP = deoxyuridine monophosphate, dTMP = thymidine monophosphate.

- ✓ **Comment.** As for the 19-F assay, the authors should add a broader discussion of previous uses of 19-F NMR to study PLP enzyme inhibition. Some references for these discussion points are suggested below:

Recent reference on dihydrofolate reductase DHFR inhibition for anti-cancer therapeutics development:

Ng, Hui-Li; Ma, Xiang; Chew, Eng-Hui; Chui, Wai-Keung “Design, synthesis, and biological evaluation of coupled bioactive scaffolds as potential anticancer agents for dual targeting of dihydrofolate reductase and thioredoxin reductase,” *J. Med. Chem.* **2017**, *60*, 1734-1745.

Recent references on thymidylate synthase (TS) inhibition for anti-cancer therapeutics development:

Negrei, Carolina; Hudita, Ariana; Ginghina, Octav; Galateanu, Bianca; Voicu, Sorina Nicoleta; Stan, Miriana; Costache, Marieta; Fenga, Concettina; Drakoulis, Nikolaos; Tsatsakis, Aristidis M. “Colon cancer cells gene expression signature as response to 5- fluorouracil, oxalipatin, and folinic acid treatment,”

Frontiers in Pharmacology **2016**, *7*, 172/1-172/10

Azijli, Kaamar; van Roosmalen, Ingrid A. M.; Smit, Jorn; Pillai, Saravanan; Fukushima, Masakazu; de Jong, Steven; Peters, Godefridus J.; Bijnsdorp, Irene V.; Kruyt, Frank A. E. “The novel thymidylate synthase inhibitor trifluorothymidine (TFT) and TRAIL synergistically eradicate non-small cell lung cancer cells.”

Cancer Chemotherapy and Pharmacology **2014**, *73*, 1273-1283.

Gomez-Martin, C.; Salazar, R.; Montagut, C.; Gil-Martin, M.; Nunez, J. A.; Puig, M.; Lin, X.; Khosravan, R.; Tursi, J. M.; Lechuga, M. J.; et al. “A phase I, dose-finding study of sunitinib combined with cisplatin and 5-fluorouracil in patients with advanced gastric cancer,”

Investigational New Drugs **2013**, *31*, 390-398.

Discussions of the use of fluorinated probes for PLP enzyme activity:

Berkowitz, David B.; Karukurichi, Kannan R.; Salud-Bea, Roberto; Nelson, David L.; McCune, Christopher D. “Use of Fluorinated Functionality in Enzyme Inhibitor Development – Mechanistic and Analytical Advantages” *J. Fluor. Chem.* **2008**, *129*, 731- 742.

Karukurichi, Kannan R.; Salud-Bea, Roberto; Jahng, Wan Jin; Berkowitz, David B. “Examination of the New alpha-(2’Z-Fluoro)vinyl Trigger with Lysine Decarboxylase: The Absolute Stereochemistry Determines the Reaction Course,” *J. Am. Chem. Soc.* **2007**, *129*, 258-259.

>>>Response to Comment.

We thank the reviewer for the comment to improve this manuscript. Following the reviewer's comment, we have added new sentences about the previous uses of ^{19}F NMR, with appropriate references.

[Revised manuscript, p.10, lines 5–10]

NMR is also a remarkable modality, which has a high signal transparency even in opaque samples. Because of a good transparency of NMR signals, NMR molecular probes are suitable for the analysis under crude biological conditions. ^{19}F NMR, which is a nucleus with high sensitivity next to ^1H , has been actively used in the design of NMR molecular probes for the analysis of biomolecules.^{30–37}

Reviewer #2

Original comments from Reviewer #2

Reviewer #2 (Remarks to the Author):

The manuscript under consideration by Sando and co-workers describes the development of fluorescent and ^{19}F probes for serine hydroxymethyltransferase (SHMT) activity. The general design strategy exploits the retro-aldol-type reaction to generate an aldehyde product. In the case of the fluorescent probe the authors selected a N,N-dialkyl naphthalene dye which upon reaction with SHMT generates an aldehyde that is in conjugation with the electron donating moiety. This leads to a change in the fluorescent properties of the probe. Likewise, the ^{19}F probe relies shifting the fluorine resonance upon reaction, a strategy introduced by Ralph Mason and co-workers over a decade ago (10.1016/j.mri.2006.04.003). The authors claim their probes hold distinct advantages over existing methods such as coupled enzyme assays. The authors also noted that their probes contain two asymmetric centers and thus motivated them to synthesize the various isomers to elucidate which form would be the optimal substrate for SHMT. The author complimented these studies by determining the rates for each isomer. The author then performed NMR / MRI studies using the ^{19}F probe and showed a difference in signal when comparing the unactivated probe and the SHMT-treated probe. Finally, the authors employed their fluorescent probe in a screen to identify 2 inhibitors of SHMT activity. Overall this study was nice but as a reviewer I do not believe it is at the caliber of Nature Communications. The authors are urged to consider resubmitting to a different journal after addressing the points below.

- 1) One of the major advantages of having a direct SHMT probe over existing methods such as the coupled-enzyme assay is that it can be used on intact samples (e.g., live cells). However, in each of the instances the authors selected to use the probe in vitro or in homogenates. I suspect the issue is that the probes are not cell permeable and thus, the authors were constrained to working around this limitation. If my suspicions are correct the authors should mask the carboxylate with a chemical group such as AM ester which can then be removed by intracellular esterases.
- 2) The authors mentioned there are two SHMT isoforms, why was this only tested against only one of the isoforms? How can you ensure that you are not getting pan-reactivity? And if you do have reactivity with both (which should be the case based on the proposed

mechanism) this decreases the utility of the probe because in each instance you will need to do much more elaborate control experiments with siRNA KO etc.

3) The authors were not careful in characterizing the stability of the probe or of the turned over product. This was a major sticking point for me. In particular, are there enzymes found in living systems can activate or change the properties of the probe? The authors should at least test this against CYPs. More concerning, however, is the stability of the turned over aldehyde products. The authors made no attempts to evaluate its stability after the product has been formed. Fluorogenic aldehyde probes can form Schiff-bases, iminiums, they can cyclize with Cys and homo-Cys (see aldehyde based probes for these amino acids) and they can be oxidized to the acid product by a variety of ROSs. Since the authors claim their probe will likely find utility in cancer, oxidative stress in tumors is a major problem.

4) When I review papers for high profile journals such as Nat Commun, I consider whether the paper can spark new discovery in various fronts. From a probe perspective the current design is limited to only producing aldehyde products. The major problem with this is that aromatic aldehydes are typically non-fluorescent due to donor-PeT quenching. This means any other dye scaffold other than the naphthalene used by the authors would result in an on-off response. Turn-off probes are not desirable in the community because other factors can lead to a decrease in signal such as dye efflux, bleaching etc. In fact, this is likely why the authors decided to make a ^{19}F probe instead of other fluorescent analogs because the MRI version is not impacted by quenching. I thought this was clever and resourceful.

5) While it may seem on first glance that the authors were being scientifically rigorous by determine which isomer is the best substrate. These experiments are simply not important, nor do add significantly to the paper. If this information was responsible for helping the authors design the current probe that is a different story. As is, it is just distracting. It is similar to how a lot of probe-based papers add DFT calculations that have no true value beyond bulking up the study.

6) Developing ^{19}F MRI probes is an area that our group has strongly considered in the past. However, there were several important reasons that ultimately caused us to not pursue this area. Firstly, ^{19}F is not used clinically and it does not seem like this will happen in the future. The issue is the poor sensitivity. If the authors compare the ^1H MRI images with their ^{19}F images it is abundantly clear this is the case. Most MRI facilities do not have coils that can detect ^{19}F which means this is really a niche area. Lastly, having talked to a lot of MRI experts in the field they all told us that dynamic probes is 'neat and cool' by not

desirable for real translational applications. Since you already have an issue seeing any signal at, imagine if 20% of that signal is shifted to another species. How are you going to see that?

7) I understand it was necessary to do something more with the probe to try make the study more comprehensive. I would have preferred the cell studies mentioned above but because it does not seem like the probes are compatible with live systems, the authors decided to perform a screen. You can do exactly the same screen using the coupled enzyme assay since it is in vitro. While the coupled assay may be operationally more complex, it can still be done. This brings us back to the question of whether the current probes have made a big enough of an advance to warrant publication in a high profile journal.

Point-by-point response to the comments of Reviewer #2

We wish to thank the reviewer for the comments and for providing constructive suggestions. We have addressed all the points raised by the reviewer through discussions and/or new experiments.

- ✓ **Comment.** 1) One of the major advantages of having a direct SHMT probe over existing methods such as the coupled-enzyme assay is that it can be used on intact samples (e.g., live cells). However, in each of the instances the authors selected to use the probe in vitro or in homogenates. I suspect the issue is that the probes are not cell permeable and thus, the authors were constrained to working around this limitation. If my suspicions are correct the authors should mask the carboxylate with a chemical group such as AM ester which can then be removed by intracellular esterases.

>>>Response to Comment.

We thank the reviewer for the constructive comment. We agree that it is important to demonstrate the biological applications of our probes on intact samples, e.g., live cells. However, the main purpose of this research is to demonstrate a general strategy to design SHMT chemical probes, which had not been realized, and to find new inhibitor candidates against SHMT.

As newly explained in an introduction section (Revised manuscript, p.4, line 13 – p.5, line 3), the development of SHMT inhibitors has attracting attentions, because these have potential for treatment of two types of important diseases. First one is antimalarial drug. Malaria is a life-threatening disease that spreads to people through infected anopheles mosquitoes. It has a tremendous impact globally, 216 million people are infected in 2016, and 445,000 people have died. In addition, the resistance of malaria parasites against existing antimalarial drugs has become a serious problem. Under such circumstances, researches and investigations for new inhibitors against malaria SHMT have been conducted, and SHMT is drawing attention as a promising target enzyme. Second one is anticancer drug. In chemotherapy, three enzymes of one-carbon metabolism; SHMT, dihydrofolate reductase (DHFR), and

thymidylate synthase (TS), are potent target enzymes strongly related to cell proliferation. Actually, inhibitors targeting DHFR and TS, such as methotrexate and fluorouracil respectively, have been used for a long time as important anticancer agents. Among the three enzymes of the one-carbon metabolism, to our knowledge, human SHMT is the only enzyme to which an established chemotherapeutic agent has not been developed yet. Therefore, human SHMT has attracted attention as a target enzyme of anticancer drug.

In fact, the present design strategy realized the development of the first fluorescent and ^{19}F chemical probes targeting SHMT (**Fig. 2**), which could be used for inhibitor screenings under crude/opaque biological conditions, and realized the successful finding of two new SHMT inhibitor candidates (**Fig. 6**). We believe that this is one of the most significant features of this paper, and in-cell applications are beyond this initial scope.

However, we agree with your comment that it is also important to show the possibility that the present design strategy can be used in designing fluorescent probes for various biological applications. Therefore, based on the present design strategy, we additionally demonstrated the development of a new type of fluorescence turn-on probe **S1** for SHMT, which produces fluorescent chromophore after tandem retro-aldol- β -elimination reaction (**Supplementary Fig. S13**). This result clearly indicates that our probe design can be applied for not only aldehyde-carrying DMANA but also other bright fluorophores such as hydroxycoumarin and resorufin which have been used in the design of fluorescent molecular probes for biological applications including cellular assays (*e.g.* C. J. Chang *et al.*, *Chem. Commun.*, 44, 4647–4649 (2007) and S. Q. Yao *et al.*, *Nat. Commun.* 5, 3276 (2014)). The successful development of new probe **S1** provides researchers with important choice in the design of SHMT probe and further strengthen the robust versatility of the present design strategy.

These points have been discussed in the main text (p. 23, line 18 –

p. 24, line 2) and new Supplementary Figure has been added in Supplementary information (**Supplementary Fig. S13**).

[Revised manuscript, p. 23, line 18 – p. 24, line 2]

In fact, based on the strategy, we could also develop a new type of fluorescence turn-on probe for SHMT, which produces fluorescent chromophore upon reaction with SHMT through tandem retro-aldol- β -elimination reaction (**Supplementary Fig. S13, S14**). This new turn-on mechanism could allow researchers to design SHMT probe with various bright fluorophores such as hydroxylcoumarin and resorufin,^{41,42} demonstrating the versatility of this design strategy.

Supplementary Figure S13 | β -elimination-triggered fluorescence probe targeting

SHMT. a) Schematic illustration of β -elimination-triggered SHMT fluorescence probe **S1**. b) Fluorescence spectral change of probe **S1** (1 μ M) during the SHMT1-catalyzed reaction from 0 to 60 min. Excitation at 390 nm. Assay conditions: 1 unit/mL SHMT1, 50 mM HEPES buffer (pH 7.5), 100 mM NaCl, 0.5 mM EDTA, 1 mM dithiothreitol (DTT), with or without inhibitor (\pm)-SHIN1 10 μ M, 0.2% DMSO, 37 $^{\circ}$ C. Fluorescence spectra were measured at 37 $^{\circ}$ C using a JASCO FP-6500 fluorescence spectrometer. The reaction buffer (1 unit/mL hSHMT1, 50 mM HEPES pH 7.5, 100 mM NaCl, 0.5 mM EDTA, 1 mM DTT, 0.2% DMSO) was pre-incubated at 37 $^{\circ}$ C for 1 h. The fluorescence

probe **S1** was added to the solution and incubated at 37 °C. Fluorescence emission upon excitation at 390 nm was monitored for 60 min.

- ✓ **Comment.** 2) The authors mentioned there are two SHMT isoforms, why was this only tested against only one of the isoforms? How can you ensure that you are not getting pan-reactivity? And if you do have reactivity with both (which should be the case based on the proposed mechanism) this decreases the utility of the probe because in each instance you will need to do much more elaborate control experiments with siRNA KO etc.

>>>Response to Comment.

We thank for this suggestion. According to the reviewer's comment, we have checked the SHMT isoform selectivity of our fluorescence probe. It was found that fluorescence probe **1** reacts with both isoforms SHMT 1 and 2. This pan-reactivity would be useful for SHMT researches such as high throughput inhibitor screening against SHMT2. Since SHMT is present in various species and has several isoforms, the wide-reactivity against SHMTs could be a benefit for various applications. The information about the SHMT isoform selectivity has been added in the main text (p. 14, lines 9–12) and supplementary information (**Supplementary Fig. S8**).

Supplementary Figure S8 | Fluorescence spectral change of probe 1 during the SHMT2-catalyzed reaction. Excitation at 390 nm. Assay conditions: 5.0 μ M probe **1**, 1 unit/mL SHMT2, 50 mM HEPES buffer (pH 7.5), 100 mM NaCl, 0.5 mM EDTA, 1 mM dithiothreitol (DTT), 1.0% DMSO, 37 °C. Fluorescence spectra were measured using a JASCO FP-6500 fluorescence spectrometer.

In addition, under our experimental conditions (units of SHMT were defined by the reactivity with *DL-erythro*- β -phenylserine), fluorescent probe **1** was found to be more responsive to SHMT2 than SHMT1. This fact indicates that it would be possible to design SHMT molecular probes with higher isoform selectivity. In fact, we have already obtained several SHMT substrates that show high isomer selectivity depending on the structure of chromophore. We had not include this information in this manuscript because it had been a different topic from the main purpose of this research, reporting the general molecular design and the inhibitor screening for SHMT. However, if the editor or reviewer thinks this data is better to be included in this manuscript, we will consider adding it.

- ✓ **Comment.** 3) The authors were not careful in characterizing the stability of the probe or of the turned over product. This was a major sticking point for me. In particular, are there enzymes found in living systems can activate or change the properties of the probe? The authors should at least test this against CYPs. More concerning, however, is the stability of the turned over aldehyde products. The authors made no attempts to evaluate its stability after the product has been formed. Fluorogenic aldehyde probes can form Schiff-bases, iminiums, they can cyclize with Cys and homo-Cys (see aldehyde based probes for these amino acids) and they can be oxidized to the acid product by a variety of ROSs. Since the authors claim their probe will likely find utility in cancer, oxidative stress in tumors is a major problem.

>>>Response to Comment.

We thank the reviewer for this important advice. We agree with the reviewer's comment concerning the stability of the probe and the turned over product. The stability of the probe under biological conditions was partly shown in **Fig. 5d** (^{19}F NMR in rat liver homogenate condition). The ^{19}F NMR probe worked in rat liver homogenate including various metabolizing enzymes including CYPs. Such a crude enzyme mixture did not affect the ^{19}F NMR signal of the probe and product.

Furthermore, following the reviewer's comment about the stability of the fluorescence probe **1** and the turned over product DMANA carrying aldehyde, we have additionally checked the stability of the probe and product against biological reactants: glutathione, L-cysteine, DL-homocysteine, H_2O_2 , NaOCl , KO_2 , TBHP, and NOC7 (reaction conditions are shown in the caption of **Supplementary Fig. S6**). In all conditions, no significant changes in fluorescence were observed, supporting the stability of the probe and product against these biological reactants in this time scale. These points are discussed in the main text (p. 14, lines 1–4) and new **Supplementary Fig. S6** in Supplementary Information.

Supplementary Figure S6 | The stability of a) probe 1 and b) turned over product DMANA against reactive oxygen species and reactive species having thiols and amines. a) Fluorescence intensity change of probe 1 (1 μ M). b) Fluorescence intensity change of DMANA (5 μ M). Assay conditions: Incubation time for 60 min at 37 $^{\circ}$ C. Excitation at 390 nm. 50 mM HEPES buffer (pH 7.5), 100 mM NaCl, 0.5 mM EDTA, 1 mM dithiothreitol (DTT), 1% DMSO. Fluorescence spectra were measured using a SHIMADZU RF-6000 fluorescence spectrometer (a) and using a JASCO FP-6500 fluorescence spectrometer (b). Error bars represent s.d., $n = 3$.

- ✓ **Comment.** 4) When I review papers for high profile journals such as Nat Commun, I consider whether the paper can spark new discovery in various fronts. From a probe perspective the current design is limited to only producing aldehyde products. The major problem with this is that aromatic aldehydes are typically non-fluorescent due to donor-PeT quenching. This means any other dye scaffold other than the naphthalene used by the authors would result in an on-off response. Turn-off probes are not desirable in the community because other factors can lead to a decrease in signal such as dye efflux, bleaching etc. In fact, this is likely why the authors decided to make a 19F probe instead of other

fluorescent analogs because the MRI version is not impacted by quenching. I thought this was clever and resourceful.

>>>Response to Comment.

We thank the reviewer's comment. To rebut the reviewer's comment about the limitation of our fluorescent probe design, we have newly designed a new type of turn-on fluorescence probe **S1**. The probe **S1** was synthesized by 4-steps as shown in **Supplementary Fig. S14**.

Supplementary Figure S14 | Synthesis of the SHMT fluorescence probe S1.

The new probe **S1** is designed to produce fluorescent chromophore upon reaction with SHMT through tandem retro-aldol- β -elimination reaction (**Supplementary Fig. S13a**). In fact, the new probe **S1** reacted with SHMT1 and produced fluorescent 7-hydroxycoumarin carrying no aldehyde. It was confirmed to be a turn-on type SHMT fluorescent probe (**Supplementary Fig. S13b**). This result clearly indicates that our probe design is not limited to only aromatic aldehyde and that other bright fluorophores such as hydroxycoumarin and resorufin can be used as same

as previously reported turn-on fluorescent probes based on the β -elimination mechanism, *e.g.* C. J. Chang *et al.*, *Chem. Commun.*, 44, 4647–4649 (2007) and S. Q. Yao *et al.*, *Nat. Commun.*, 5, 3276 (2014). Thanks to this revision based on the comment by reviewer, we are sure that our design strategy is further strengthened.

These points are discussed at the conclusion section in main text (p. 23, line 18 – p. 24, line 2 and **Supplementary Fig. S13**).

Supplementary Figure S13 | β -elimination-triggered fluorescence probe targeting SHMT. a) Schematic illustration of β -elimination-triggered SHMT fluorescence probe **S1**. b) Fluorescence spectral change of probe **S1** (1 μ M) during the SHMT1-catalyzed reaction from 0 to 60 min. Excitation at 390 nm. Assay conditions: 1 unit/mL SHMT1, 50 mM HEPES buffer (pH 7.5), 100 mM NaCl, 0.5 mM EDTA, 1 mM dithiothreitol (DTT), with or without inhibitor (\pm)-SHIN1 10 μ M, 0.2% DMSO, 37 $^{\circ}$ C. Fluorescence spectra were measured at 37 $^{\circ}$ C using a JASCO FP-6500 fluorescence spectrometer. The reaction buffer (1 unit/mL hSHMT1, 50 mM HEPES pH 7.5, 100 mM NaCl, 0.5 mM EDTA, 1 mM DTT, 0.2% DMSO) was preincubated at 37 $^{\circ}$ C for 1 h. The fluorescence probe **S1** was added to the solution and incubated at 37 $^{\circ}$ C. Fluorescence emission upon excitation at 390 nm was monitored for 60 min.

- ✓ **Comment.** 5) While it may seem on first glance that the authors were being scientifically rigorous by determine which isomer is the best substrate. These experiments are simply not important, nor do add significantly to the paper. If this information was responsible for helping the authors design the current probe that is a different story. As is, it is just distracting. It is similar to how a lot of probe-based papers add DFT calculations that have no true value beyond bulking up the study.

>>>Response to Comment.

Thank you for the comment. Stereoisomer selectivity was found for the first time by actually conducting experiments this time. Therefore, we thought that it was valuable data to describe in this paper. If you still think it is not necessary, we will consider deleting them.

- ✓ **Comment.** 6) Developing ^{19}F MRI probes is an area that our group has strongly considered in the past. However, they were several important reasons that ultimately caused us to no pursue this area. Firstly, ^{19}F is not used clinically and it does not seem like this will happen in the future. The issue is the poor sensitivity. If the authors compare the ^1H MRI images with their ^{19}F images it is abundantly clear this is the case. Most MRI facilities do not have coils that can detect ^{19}F which means this is really a niche area. Lastly, having talked to a lot of MRI experts in the field they all told us that dynamic probes is ‘neat and cool’ by not desirable for real translational applications. Since you already have an issue seeing any signal at, imagine if 20% of that signal is shifted to another species. How are you going to see that?

>>>Response to Comment.

We agree with the reviewer’s comment concerning the sensitivity of ^{19}F NMR/MRI. As pointed out by the reviewer, ^{19}F MRI has not been used practically. However, we believe that there is a possibility of future development in ^{19}F MRI. ^{19}F MRI could detect higher signal with ultra high field MRI. Recently, 7T MRI was approved for clinical use by FDA and 11.7 T MRI was manufactured for experimental use. With such super

high field MRI, our research will be more practical. In the future, multichannel phased array reception, sparse sampling, noise elimination technology by AI etc. would greatly improve SNR of ^{19}F MRI. Since ^{19}F MRI can share many high frequency devices with ^1H MRI, technically it can be used in clinical equipment simply by adding a receiving coil. With ^1H MRI, it is difficult to identify the lesion, and it is also true that many doctors and researchers are seeking probe/agents that are more specific to lesions and specific metabolism. Once critical imaging agents are developed, we believe there is a possibility of being used practically in laboratory and hopefully in clinic.

However, it is also true that at present it is difficult to use our ^{19}F MRI probe for *in vivo* MRI. From the viewpoint of development of molecular probes, there are several spaces where researchers can contribute. Increasing the number of ^{19}F atoms on molecular probes, in addition to further enhancing the sensitivity of MRI instrument, would be one important option. Furthermore, development of new SHMT molecular probes (^{13}C , ^{15}N , and ^{19}F) based on the current design strategy, that can be used in state-of-the-art NMR-hyperpolarization technique, may be another option to realize dramatic increase of sensitivity

To show the limitation at this stage and future direction, we added the sentence in Discussion section in main text (p. 24, lines 15–19).

[Revised manuscript, p.24, lines 15–19]

For *in vivo* applications, it would be necessary to improve the sensitivity. This may be achieved by increasing the number of ^{19}F atoms on molecular probes or further enhancing the sensitivity of MRI instrument.^{30,38} Combination with state-of-the-art NMR-hyperpolarization technique may be another option.^{43,44}

- ✓ **Comment.** 7) I understand it was necessary to do something more with the probe to try make the study more comprehensive. I would have preferred the cell studies mentioned above but because it does not seem like the probes are compatible with live systems, the authors decided to perform a screen. You can do exactly the same screen using the coupled enzyme assay since it is in vitro. While the coupled assay may be operationally more complex, it can still be done. This brings us back to the question of whether the current probes have made a big enough of an advance to warrant publication in a high profile journal.

>>>Response to Comment.

We thank the reviewer for the constructive comment. As pointed out by reviewer, it is possible to detect the activity of SHMT by a coupled assay system if it is under a purified condition. On the other hand, it is also important to evaluate the inhibitor ability under biological/opaque conditions, where many biological components co-exist, even at the screening stage. It allows us to evaluate the selectivity of inhibitors against other off-target proteins and biological components.

To strengthen the merit of our SHMT probe, we have conducted the demonstration of inhibitor screening in mouse liver homogenate by using ¹⁹F NMR probe **2** (**Supplementary Fig. S11**). Even under a such crude condition, our ¹⁹F NMR probe **2** could easily monitor the inhibition ability of inhibitor candidates. This fact clearly indicates that SHMT activity sensing method by using our probes has an advantage compared to coupled assays. To clarify this point, we have newly added the evaluation process of inhibitor performance in mouse liver homogenate to the screening process in the main text (p. 19, line 20 through p. 20, line 2, **Fig. 6a**) with new **Supplementary Fig. S11** (Supplementary information).

[Revised manuscript, p.19, line 20 through p. 20, line 2]

Finally, in order to confirm the efficacy of candidate inhibitors under crude biological conditions, we evaluated the inhibition of SHMT activity in mouse liver homogenate by using ¹⁹F NMR probe **2** (**Supplementary Fig. S11**). After the five stages of screening, two candidate compounds Hit 1 and 2 were obtained (**Fig. 6b**).

Supplementary Figure S11 | Confirmation of SHMT activity inhibition in mouse liver homogenate by using ^{19}F NMR probe 2. 50 mM HEPES pH 7.5, 100 mM NaCl, 0.5 mM EDTA, 1 mM DTT, 30% D_2O , 16.7 mg proteins/mL, 1 mM ^{19}F NMR probe 2, 1% DMSO, without (a) or with 10 μM inhibitor candidate (b, c).

Reviewer #3

Original comments from Reviewer #3

Reviewer #3 (Remarks to the Author):

Recommendation: Publish in Nature Comm. after minor revisions.

Comments:

The manuscript „Design Strategy for Serine Hydroxymethyltransferase Probes Based on Retro-Aldol-Type Reaction” from the Prof. Sando et al. describes the development of chemical probes for the enzyme SHMT as markers for the application in fluorescence and ¹⁹F NMR spectroscopy. Furthermore, they were able to demonstrate the convenience and benefit of their methods by e.g. applying the chemical probes to a high-throughput screening to identify two potential lead structures for drug development, which is a highlight of the paper.

This work nicely illustrates the use of a SHMT-induced retro-aldol reaction to access either fluorescent or ¹⁹F labelled chemical probes on a rather complex target. Especially, the ¹⁹F labelled chemical probe was shown be a potential tool for the SHMT detection in biological samples. Overall the work is of high quality and merits the publication in Nature Comm. It will find a broad audience that includes pharmacologists, medicinal chemists, and biochemists. This publication will fill a gap in the development of SHMT-related potential drugs and will contribute to a success in this field.

Major revisions:

Despite the great science in this work, the major weakness of this manuscript is the writing, which does not appear the standard of Nature Comm. and needs some improvement. Furthermore, the literature is not appropriately covered in the introduction part, which is already proven by “only” 19 citations in total.

1) The fields of fluorine MRS and MRI in biomedicine and fluorescence spectroscopy are inadequately represented missing impactful but also historically important reviews and articles (e.g. 1. Ahrens et al. "In vivo imaging platform for tracking immunotherapeutic cells." Nature Biotechnology 23: 983. 2. Ruiz-Cabello J, Barnett BP, Bottomley PA, Bulte JWM. Fluorine ¹⁹F MRS and MRI in biomedicine. NMR in Biomed. 2011;24:114-29. 3.

Ahrens ET1, Bulte JW. Tracking immune cells in vivo using magnetic resonance imaging. *Nat Rev Immunol.* 2013;13:755-63. 4. Weiss, S. "Fluorescence Spectroscopy of Single Biomolecules." *Science* 1999, 283 (5408): 1676-1683....). Those are just a couple examples but there are many more. Therefore, the introduction requires minor corrections and highlighting of the chosen methods and their advantages.

2) The research on the target SHMT does not only include the therapeutic area of cancer but also Malaria, which needs to be mentioned and introduced, because the pyrazolopyran-based inhibitors reported in the manuscript were also used against *Plasmodium falciparum* (e.g. 1. Maenpuen, S.; Sopitthummakhun, K.; Yuthavong, Y.; Chaiyen, P.; Leartsakulpanich, U. Characterization of *Plasmodium falciparum* serine hydroxymethyltransferase–A potential antimalarial target *Mol. Biochem. Parasitol.* 2009, 168, 63– 73; 2. Sopitthummakhun, K.; Maenpuen, S.; Yuthavong, Y.; Leartsakulpanich, U.; Chaiyen, P. Serine hydroxymethyltransferase from *Plasmodium vivax* serine is different in substrate specificity from its homologues *FEBS J.* 2009, 276, 4023– 4036....). The introduction might cover more information about the target SHMT and the approaches to address this target.

3) The entire manuscript requires some harmonization and standardization. (examples are given below).

4) The authors could comment on possible side-effects and toxicity of the corresponding aldehydes especially Schiff-base formation in vivo.

5) Please further elucidate the homology especially of the active site between the three species. How adaptable is the mouse model to humans?

6) Please comment on the feasibility of the compounds for further use in in vivo experiments. Is imaging in mice/rats possible?

7) The modelling process needs some explanation. Was there a program used or were the ligands manually placed? Otherwise this computer-supported experiment needs to be made to prove the hypothesis of the size-dependent kinetic results.

Minor revisions:

Page 4, Caption 1: ...SHMT. a) Serine-glycine ... THF. b) Schematic ...

Page 4, Caption 1: Red circle highlights the carbon ...

Page 4, Caption 1: two abbreviations for 5,10-methylene-tetrahydrofolate (CH₂-THF and 5,10-CH₂-THF), please keep consistency.

Page 4, Caption 1: upper case for positive charge of NADP+.

Page 6, line 6: Figure 2a shows ...

Page 6, line 10: Figure 2b shows ... (redundant to the previous section and bumpy to read).

Page 6, line 10: The sentence “When SHMT ...to afford glycine” requires rewriting.

Page 6, line 13: “crowded” colloquial writing

Page 6, line 17: “When considering...” on the one hand redundant and on the other stylistically poor written

Page 6, line 18: The sentences on this page require rewriting.

Page 7, Figure 2: ...probes. a Substrate... ...b ... c... (these minor errors needs to be corrected in the entire manuscript).

Page 7, Figure 2: PDB ID: 1EJI in capital letters

Page 7, Figure 2 : Oxygen: red ; nitrogen: blue ...

Page 9, Figure 3 ; From a chemist point of view, the scheme should contain the temperatures, rxt times, and the deprotection conditions in detail, or should at least be mentioned in the caption of the Figure.

Page 9, Figure 3: Oxygen: red; nitrogen: blue...

Page 10, line 15: “naked eye” colloquial writing

Page 10, line 17-21: redundant and very difficult to read.

Page 10, line 22: “The kinetic parameters ...” sentence needs to be rewritten

Page 11, Figure 4: see comments above: a... b...

Page 14, Figure 5: Misspelling of “inhibitor” in e) (see comments above)

Page 15, line 17: ...selectivity by a counter assay

Page 15, line 21: ...Hit 1 and 2...

Page 16, line 23: ...both of the Hit compounds...

Page 17, Figure 6: b) (Top) ... and (bottom) ...

Page 19, line 2: We have succeeded, for the first time, the design of SHMT molecular probes by focusing...

Page 19, line 11: “...SHMT has been thought to be difficult.” Sentence sounds colloquial, a better sentence might be: ...considered as a challenging goal.

Point-by-point response to the comments of Reviewer #3

We wish to thank the reviewer for the comments and for providing constructive comments and comprehensive English proofreading. We have addressed all the points raised by the reviewer through new experiments and/or new text.

- ✓ **Comment.** Despite the great science in this work, the major weakness of this manuscript is the writing, which does not appear the standard of Nature Comm. and needs some improvement. Furthermore, the literature is not appropriately covered in the introduction part, which is already proven by “only” 19 citations in total.

>>>Response to Comment.

We thank the reviewer for the constructive comment to improve this manuscript. Following the reviewer’s comment, we have revised the manuscript and added 25 references for covering related literatures appropriately. We hope that our revisions have adequately addressed the reviewer’s concerns.

- ✓ **Comment.** 1) The fields of fluorine MRS and MRI in biomedicine and fluorescence spectroscopy are inadequately represented missing impactful but also historically important reviews and articles (e.g. 1. Ahrens et al. "In vivo imaging platform for tracking immunotherapeutic cells." Nature Biotechnology 23: 983. 2. Ruiz-Cabello J, Barnett BP, Bottomley PA, Bulte JWM. Fluorine 19F MRS and MRI in biomedicine. NMR in Biomed. 2011;24:114-29. 3. Ahrens ET1, Bulte JW. Tracking immune cells in vivo using magnetic resonance imaging. Nat Rev Immunol. 2013;13:755-63. 4. Weiss, S. "Fluorescence Spectroscopy of Single Biomolecules." Science 1999, 283 (5408): 1676-1683....). Those are just a couple examples but there are many more. Therefore, the introduction requires minor corrections and highlighting of the chosen methods and their advantages.

>>>Response to Comment.

We thank the reviewer for the constructive advice. Following the reviewer's comment, we have added new sentences about the advantage and precedents of ^{19}F NMR/MRI, with appropriate references (Revised manuscript, p.10, lines 5–10). We hope that our revisions have adequately addressed the reviewer's concerns.

- ✓ **Comment.** 2) The research on the target SHMT does not only include the therapeutic area of cancer but also Malaria, which needs to be mentioned and introduced, because the pyrazolopyran-based inhibitors reported in the manuscript were also used against *Plasmodium falciparum* (e.g. 1. Maenpuen, S.; Sopitthummakhun, K.; Yuthavong, Y.; Chaiyen, P.; Leartsakulpanich, U. Characterization of *Plasmodium falciparum* serine hydroxymethyltransferase—A potential antimalarial target *Mol. Biochem. Parasitol.* 2009, 168, 63– 73; 2. Sopitthummakhun, K.; Maenpuen, S.; Yuthavong, Y.; Leartsakulpanich, U.; Chaiyen, P. Serine hydroxymethyltransferase from *Plasmodium vivax* serine is different in substrate specificity from its homologues *FEBS J.* 2009, 276, 4023–4036....). The introduction might cover more information about the target SHMT and the approaches to address this target.

>>>Response to Comment.

Thank you for your valuable comment to improve this manuscript. Following the comments, we have added sentences to explain the importance of SHMT in the therapeutic area of Malaria and the previous efforts of anti-Malaria drug to the manuscript.

[Revised manuscript, p.4, lines 13–19]

The development of SHMT inhibitors has been performed especially toward treatment of two types of diseases. The first one is antimalarial drug.^{8–15} Malaria is a life-threatening disease that spreads to people through infected anopheles mosquitoes. It has a tremendous impact globally, 216 million people are infected in 2016, and 445,000 people have died.¹³ In addition, the resistance of malaria parasites against

existing antimalarial drugs has become a serious problem. Under such circumstances, researches and investigations for new inhibitors against malaria SHMT has been conducted.

- ✓ **Comment.** 3) The entire manuscript requires some harmonization and standardization. (examples are given below).

>>>Response to Comment.

We thank the reviewer for the constructive comment. Following the reviewer's comment, we have revised the manuscript. We hope that our revisions have adequately addressed the reviewer's concerns. If the reviewer needs more rewriting, we are ready to rewrite again.

- ✓ **Comment.** 4) The authors could comment on possible side-effects and toxicity of the corresponding aldehydes especially Schiff-base formation in vivo.

>>>Response to Comment.

We agree with the reviewer's comment. The product aldehyde might react with biological relevant nucleophiles through forming Schiff's base. Following the reviewer's comment, first we have checked the stability of fluorescence product aldehyde (DMANA) against biological reactants, *e.g.* glutathione, L-cysteine, DL-homocysteine, H₂O₂, NaOCl, KO₂, TBHP, and NOC7 (**Supplementary Fig. S6**). Under all conditions, no significant changes in fluorescence were observed.

Furthermore, we have evaluated the toxicity of DMANA (**Supplementary Fig. S7**). Significant toxicity was not observed from the cytotoxicity test. These results indicate that the significant side-effects and toxicity are not problematic under the concentration and time range of fluorescence experiments. We have added the discussion about the stability and toxicity of DMANA in the main text and Supplementary Information (p. 14, lines 1–8, **Supplementary Fig. S6, Supplementary Fig. S7**).

However, under the concentration range of ¹⁹F NMR/MRI, ¹⁹F product aldehyde might show the side-effects and toxicity, because the required concentration is high for detection of ¹⁹F NMR/MRI signal. For practical *in vivo* MRI, it might be necessary to improve the sensitivity to lower the required concentration. These points have also been discussed in the main text (p. 24, lines 15–19).

Supplementary Figure S6 | The stability of a) probe 1 and b) turned over product DMANA against reactive oxygen species and reactive species having thiols and amines. a) Fluorescence intensity change of probe 1 (1 μ M). b) Fluorescence intensity change of DMANA (5 μ M). Assay conditions: Incubation time for 60 min at 37 $^{\circ}$ C. Excitation at 390 nm. 50 mM HEPES buffer (pH 7.5), 100 mM NaCl, 0.5 mM EDTA, 1 mM dithiothreitol (DTT), 1% DMSO. Fluorescence spectra were measured using a SHIMADZU RF-6000 fluorescence spectrometer (a) and using a JASCO FP-6500 fluorescence spectrometer (b). Error bars represent s.d., $n = 3$.

Supplementary Figure S7 | Cytotoxicity evaluation of turned over fluorescence product DMANA. HeLa cells were plated in a 6-well plate at a density of 50,000 cells/mL in DMEM media. After incubation for 24 h, each samples were supplemented

with either 1 μM or 5 μM DMANA, containing 1% DMSO. Equivalent samples were supplemented with 1% DMSO as a vehicle control. At 6 and 24 hours, cells were dissociated from wells by trypsin, a 10 μL sample was removed from each of the samples and mixed 1:1 with a 0.4% wt/volume trypan blue solution in PBS. Samples were incubated for 1 minute at room temperature before being loaded onto a hemocytometer where live and dead cells were counted. Each sample was made in triplicate for each time point. Error bars represent s.d., $n = 3$.

- ✓ **Comment.** 5) Please further elucidate the homology especially of the active site between the three species. How adaptable is the mouse model to humans?

>>>Response to Comment.

Human SHMT shares 91% sequence identity with mouse SHMT and 42% sequence identity with *P. vivax* SHMT (PvSHMT) (Chaiyen *et al.*, *FEBS J.*, 276, 4023–4036 (2009)). Because human and mouse have a high homology, mouse SHMT is considered to be appropriate for this modeling study. We have added this point about homology in the main text (P8, lines 5–8).

- ✓ **Comment.** 6) Please comment on the feasibility of the compounds for further use in in vivo experiments. Is imaging in mice/rats possible?

>>>Response to Comment.

Basically, we think that it is difficult to conduct *in vivo* experiments by using the current probes and machine setup. However, regarding fluorescence probe **1**, depending on the two-photon efficiency of the turned-over product, it might be possible to detect the product near the surface of the mouse/rat by using a two-photon excitation microscope.

Regarding practical *in vivo* ^{19}F MRI using probe **2**, there is a problem of sensitivity, so it would be necessary to enhance the sensitivity, for example, by increasing the number of ^{19}F atoms on molecular probes or by improving NMR/MRI instruments. In order to clarify this point, we have added this discussion about current limitation and future direction in an application for ^{19}F MRI in main text. (P24, lines 15–19)

- ✓ **Comment.** 7) The modelling process needs some explanation. Was there a program used or were the ligands manually placed? Otherwise this computer-supported experiment needs to be made to prove the hypothesis of the size-dependent kinetic results.

>>>Response to Comment.

We thank the reviewer for the comment. Following the reviewer's comment, we have added an explanation of manual modeling process as follows (caption in **Supplementary Fig. S3**). We hope that our revisions have adequately addressed the reviewer's concerns.

[Revised Supplementary Information, p.S5]

Supplementary Figure S3 | Illustration of probe 1 manually fitted in active site of mSHMT1 (PDB ID: 1EJI). First, the calculation of global minimum energy conformation of *L-erythro* probe **1** was performed by Spartan'16 (Wavefunction, Inc.) software. Then, the co-crystal structure of PLP-Gly-5-CHO-THF-mSHMT1 (PDB ID: 1EJI) and probe **1** are displayed using the PyMOL software, and 5-CHO-THF is deleted. Finally, *L-erythro* probe **1** was manually placed to the active site of mSHMT1 with PLP-Gly as amino groups and carboxylic groups of Gly and probe **1**

were each superimposed. This supports the hypothesis that *L-erythro* form can be accommodated in the substrate pocket. Color code of stick model: oxygen: red; nitrogen: blue; carbon: green; PLP: orange.

- ✓ **Comment.** Minor revisions:

Page 4, Caption 1: ...SHMT. a) Serine-glycine ... THF. b) Schematic ...

>>>Response to Comment.

We thank the reviewer for the constructive comment. Following the reviewer's comment, we have changed.

- ✓ **Comment.** Page 4, Caption 1: Red circle highlights the carbon ...

>>>Response to Comment.

We thank the reviewer for the constructive comment. Following the reviewer's comments, we have changed to "highlights".

- ✓ **Comment.** Page 4, Caption 1: two abbreviations for 5,10-methylene-tetrahydrofolate (CH₂-THF and 5,10-CH₂-THF), please keep consistency.

>>>Response to Comment.

We thank the reviewer for the constructive comment. Following the reviewer's comments, we have unified to "CH₂-THF".

- ✓ **Comment.** Page 4, Caption 1: upper case for positive charge of NADP⁺.

>>>Response to Comment.

We thank the reviewer for the constructive comment. Following the reviewer's comments, we have changed positive charge to superscript "NADP⁺".

- ✓ Comment. Page 6, line 6: Figure 2a shows ...

>>>Response to Comment.

We thank the reviewer for the constructive comment. Following the reviewer's comments, we have changed.

- ✓ Comment. Page 6, line 10: Figure 2b shows ... (redundant to the previous section and bumpy to read).

>>>Response to Comment.

We thank the reviewer for the constructive comment. In the revised manuscript, we have rewritten the sentences. We hope that our revisions have adequately addressed the reviewer's concerns. If the reviewer needs more rewriting, we are ready to rewrite again.

[Revised manuscript, p.8, lines 9–15]

Figure 2a shows the crystal structure of the (5-CHO-THF)–(Gly-PLP)–SHMT ternary complex of mouse SHMT1, which has high homology to human and rat SHMT (**Fig. 2a**).²¹ Here, 5-CHO-THF plays a role as an analogue of THF in the intermediate state. In the case of THF-dependent pathway (R = H; upper arrow), SHMT transfers one carbon to THF from Ser-PLP complex, to afford glycine (**Fig. 2b**).²³ As shown in **Fig. 2a**, the vicinity of the serine recognition site is very limited. This limited space of the substrate binding site hampers the development of SHMT-responsive probes.

- ✓ **Comment.** Page 6, line 10: The sentence “When SHMT ...to afford glycine” requires rewriting.

>>>Response to Comment.

We thank the reviewer for the constructive comment. In the revised manuscript, we have rewritten the sentence. We hope that our revisions have adequately addressed the reviewer’s concerns.

- ✓ **Comment.** Page 6, line 13: “crowded” colloquial writing

>>>Response to Comment.

We thank the reviewer for the constructive comment. Following the reviewer’s comment, we have changed “crowded” to “limited”.

- ✓ **Comment.** Page 6, line 17: “When considering...” on the one hand redundant and on the other stylistically poor written

>>>Response to Comment.

We thank the reviewer for the constructive comment. In the revised manuscript, we have rewritten the sentence. We hope that our revisions have adequately addressed the reviewer’s concerns.

[Revised manuscript, p.8, lines 19–20]

In other words, using THF-independent pathway, SHMT has substrate acceptance for β -substitution of serine.

- ✓ **Comment.** Page 6, line 18: The sentences on this page require rewriting.

>>>Response to Comment.

We thank the reviewer for the constructive comment. In the revised manuscript, we have rewritten the sentences.

[Revised manuscript, p.8, lines 19–20]

In other words, using THF-independent pathway, SHMT has substrate acceptance for β -substitution of serine.

- ✓ **Comment.** Page 7, Figure 2: ...probes. a Substrate... ...b ... c... (these minor errors needs to be corrected in the entire manuscript).

>>>Response to Comment.

We thank the reviewer for the constructive comment. Following the reviewer's comment, we have changed.

- ✓ **Comment.** Page 7, Figure 2: PDB ID: 1EJI in capital letters

>>>Response to Comment.

Following the reviewer's comment, we have changed.

- ✓ **Comment.** Page 7, Figure 2 : Oxygen: red ; nitrogen: blue ...

>>>Response to Comment.

Following the reviewer's comment, we have changed.

- ✓ **Comment.** Page 9, Figure 3 ; From a chemist point of view, the scheme should contain the temperatures, rxt times, and the deprotection conditions in detail, or should at least be mentioned in the caption of the Figure.

>>>Response to Comment.

We thank the reviewer for the constructive comment. Following the reviewer's comment, we have revised the figure and caption.

- ✓ **Comment.** Page 9, Figure 3: Oxygen: red; nitrogen: blue...

>>>Response to Comment.

Following the reviewer's comment, we have changed.

- ✓ **Comment.** Page 10, line 15: “naked eye” colloquial writing

>>>Response to Comment.

We thank the reviewer for the constructive comment. Following the reviewer's comment, we have changed “naked eye” to “unaided human eye”.

- ✓ **Comment.** Page 10, line 17-21: redundant and very difficult to read.

>>>Response to Comment.

We thank the reviewer for the constructive comment. Following the reviewer's comment, we have changed as follows. We hope that our revisions have adequately addressed the reviewer's concerns.

[Revised manuscript, p.13, lines 17–19]

Upon comparing the reaction rates toward SHMT1 among the DL-*erythro* form, the DL-*threo* form, and the L-*erythro* form, it emerged that the L-*erythro* enantiomer reacted faster.

- ✓ **Comment.** Page 10, line 22: “The kinetic parameters ...” sentence needs to be rewritten

>>>Response to Comment.

We thank the reviewer for the constructive comment. Following the reviewer's comment, we have changed as follows.

[Revised manuscript, p.13, lines 20–22]

The kinetic parameters of fluorescence probe **1** (*L-erythro*) toward SHMT1 were determined to be $K_m = 1.81 \pm 0.19$ mM and $k_{cat} = 0.118 \pm 0.012$ s⁻¹.

- ✓ **Comment.** Page 11, Figure 4: see comments above: a... b...
>>>Response to Comment.
Following the reviewer's comment, we have changed.

- ✓ **Comment.** Page 14, Figure 5: Misspelling of "inhibitor" in e) (see comments above)
>>>Response to Comment.
We thank the reviewer for the constructive comment. Following the reviewer's comment, we have changed the figure and caption.

- ✓ **Comment.** Page 15, line 17: ...selectivity by a counter assay
>>>Response to Comment.
Following the reviewer's comment, we have changed.

- ✓ **Comment.** Page 15, line 21: ...Hit 1 and 2...
>>>Response to Comment.
We thank the reviewer for the constructive comment. Following the reviewer's comment, we have changed.

- ✓ **Comment.** Page 16, line 23: ...both of the Hit compounds...

>>>Response to Comment.

We thank the reviewer for the constructive comment. Following the reviewer's comment, we have changed.

- ✓ **Comment.** Page 17, Figure 6: b) (Top) ... and (bottom) ...

>>>Response to Comment.

We thank the reviewer for the constructive comment. Following the reviewer's comment, we have changed.

- ✓ **Comment.** Page 19, line 2: We have succeeded, for the first time, the design of SHMT molecular probes by focusing...

>>>Response to Comment.

We appreciate the reviewer's indication. This sentence has been corrected in the revised manuscript.

- ✓ **Comment.** Page 19, line 11: "...SHMT has been thought to be difficult." Sentence sounds colloquial, a better sentence might be: ...considered as a challenging goal.

>>>Response to Comment.

We appreciate the reviewer's indication. This sentence has been corrected in the revised manuscript.

Reviewers' comments:

Reviewer #1 (Remarks to the Author):

The authors have addressed the comments made and have improved the manuscript significantly. The revised text does need some rewording, however. Suggestions are made in the new passages - see passages highlighted in turquoise color in the attached pdf.

Reviewer #2 (Remarks to the Author):

This reviewer thanks the authors for their response. I am almost satisfied. I would still like the authors to evaluate the stability of their probe(s) against CYPs. In particular the authors can use newly purchased or prepared rat liver microsomes for these experiments.

Once this key experiment has been completed I would be supportive of publication.

Reviewer #3 (Remarks to the Author):

General response to the corrections

I would like to thank the authors for the careful and thorough corrections of the manuscript and for taking the addressed recommendations into consideration to improve the quality of the manuscript. The manuscript "Design Strategy for Serine Hydroxymethyltransferase Probes Based on Retro-Aldol-Type Reaction" from the Prof. Sando et al. fulfills the publication requirements for Nature Comm. with minor revisions.

Point to Point Response

>>>Author response.

We thank the reviewer for the constructive comment to improve this manuscript. Following the reviewer's comment, we have revised the manuscript and added 25 references for covering related literatures appropriately. We hope that our revisions have adequately addressed the reviewer's concerns.

>>>Referee response

Thank you for adding additional literature to the manuscript to cover the field.

>>>Author response.

We thank the reviewer for the constructive advice. Following the reviewer's comment, we have added new sentences about the advantage and precedents of ^{19}F NMR/MRI, with appropriate references (Revised manuscript, p.10, lines 5–10). We hope that our revisions have adequately addressed the reviewer's concerns.

>>>Referee response

Thank you for the elaboration of ^{19}F NMR techniques.

>>>Author response.

Thank you for your valuable comment to improve this manuscript. Following the comments, we have added sentences to explain the importance of SHMT in the therapeutic area of Malaria and the previous efforts of anti-Malaria drug to the manuscript. [Revised manuscript, p.4, lines 13–19] The development of SHMT inhibitors has been performed especially toward treatment of two types of diseases. The first one is antimalarial drug. 8–15 Malaria is a life-threatening disease that spreads to people through infected anopheles mosquitoes. It has a tremendous impact globally, 216 million people are infected in 2016, and 445,000 people have died.¹³ In addition, the

resistance of malaria parasites against existing antimalarial drugs has become a serious problem. Under such circumstances, researches and investigations for new inhibitors against malaria SHMT has been conducted.

>>>Referee response

I appreciate that malarial SHMT has been further introduced to the reader. But nevertheless, please rewrite this paragraph to underline the interesting and strong content with an adequate writing.

>>>Author response.

We thank the reviewer for the constructive comment. Following the reviewer's comment, we have revised the manuscript. We hope that our revisions have adequately addressed the reviewer's concerns. If the reviewer needs more rewriting, we are ready to rewrite again.

>>>Referee response

Please rewrite the introduction. It is hard to read and not harmonized yet. No changes has been conducted compared to previous version.

>>>Author response.

We agree with the reviewer's comment. The product aldehyde might react with biological relevant nucleophiles through forming Schiff's base. Following the reviewer's comment, first we have checked the stability of fluorescence product aldehyde (DMANA) against biological reactants, e.g. glutathione, L-cysteine, DL-homocysteine, H₂O₂, NaOCl, KO₂, TBHP, and NOC7 (Supplementary Fig. S6). Under all conditions, no significant changes in fluorescence were observed. Furthermore, we have evaluated the toxicity of DMANA (Supplementary Fig. S7). Significant toxicity was not observed from the cytotoxicity test. These results indicate that the significant side-effects and toxicity are not problematic under the concentration and time range of fluorescence experiments. We have added the discussion about the stability and toxicity of DMANA in the main text and Supplementary Information (p. 14, lines 1–8, Supplementary Fig. S6, Supplementary Fig. S7). However, under the concentration range of 19F NMR/MRI, 19F product aldehyde might show the side-effects and toxicity, because the required concentration is high for detection of 19F NMR/MRI signal. For practical in vivo MRI, it might be necessary to improve the sensitivity to lower the required concentration. These points have also been discussed in the main text (p. 24, lines 15–19).

Supplementary Figure S6 | The stability of a) probe 1 and b) turned over product DMANA against reactive oxygen species and reactive species having thiols and amines. a) Fluorescence intensity change of probe 1 (1 μM). b) Fluorescence intensity change of DMANA (5 μM). Assay conditions: Incubation time for 60 min at 37 °C. Excitation at 390 nm. 50 mM HEPES buffer (pH 7.5), 100 mM NaCl, 0.5 mM EDTA, 1 mM dithiothreitol (DTT), 1% DMSO. Fluorescence spectra were measured using a SHIMADZU RF-6000 fluorescence spectrometer (a) and using a JASCO FP-6500 fluorescence spectrometer (b). Error bars represent s.d., n = 3.

Supplementary Figure S7 | Cytotoxicity evaluation of turned over fluorescence product DMANA. HeLa cells were plated in a 6-well plate at a density of 50,000 cells/mL in DMEM media. After incubation for 24 h, each samples were supplemented with either 1 μM or 5 μM DMANA, containing 1% DMSO. Equivalent samples were supplemented with 1% DMSO as a vehicle control. At 6 and 24 hours, cells were dissociated from wells by trypsin, a 10 μL sample was removed from each of the samples and mixed 1:1 with a 0.4% wt/volume trypan blue solution in PBS. Samples were incubated for 1 minute at room temperature before being loaded onto a hemocytometer where live and dead cells were counted. Each sample was made in triplicate for each time point. Error bars represent s.d., n = 3.

>>>Referee response

I appreciate the effort to answer my question about stability and toxicology. The question is very well addressed, which further improves the quality of the manuscript.

>>>Author response.

Human SHMT shares 91% sequence identity with mouse SHMT and 42% sequence identity with *P. vivax* SHMT (PvSHMT) (Chaiyen et al., FEBS J., 276, 4023–4036 (2009)). Because human and mouse have a high homology, mouse SHMT is considered to be appropriate for this modeling study. We have added this point about homology in the main text (P8, lines 5–8).

>>>Referee response

Thank you for further elucidation of the homology of the species. Nevertheless, a short comment about the homology of the active site would be appropriate.

>>>Author response.

Basically, we think that it is difficult to conduct in vivo experiments by using the current probes and machine setup. However, regarding fluorescence probe 1, depending on the two-photon efficiency of the turned-over product, it might be possible to detect the product near the surface of the mouse/rat by using a two-photon excitation microscope. Regarding practical in vivo ¹⁹F MRI using probe 2, there is a problem of sensitivity, so it would be necessary to enhance the sensitivity, for example, by increasing the number of ¹⁹F atoms on molecular probes or by improving NMR/MRI instruments. In order to clarify this point, we have added this discussion about current limitation and future direction in an application for ¹⁹F MRI in main text. (P24, lines 15–19)

>>>Referee response

As this point was not only mentioned by me, it was very necessary to clarify the concerns. I appreciate the additional comment.

>>>Author response.

We thank the reviewer for the comment. Following the reviewer's comment, we have added an explanation of manual modeling process as follows (caption in Supplementary Fig. S3). We hope that our revisions have adequately addressed the reviewer's concerns. [Revised Supplementary Information, p.S5] Supplementary Figure S3 | Illustration of probe 1 manually fitted in active site of mSHMT1 (PDB ID: 1EJI). First, the calculation of global minimum energy conformation of L-erythro probe 1 was performed by Spartan'16 (Wavefunction, Inc.) software. Then, the co-crystal structure of PLP-Gly-5-CHO-THF-mSHMT1 (PDB ID: 1EJI) and probe 1 are displayed using the PyMOL software, and 5-CHO-THF is deleted. Finally, L-erythro probe 1 was manually placed to the active site of mSHMT1 with PLP-Gly as amino groups and carboxylic groups of Gly and probe 1 were each superimposed. This supports the hypothesis that L-erythro form can be accommodated in the substrate pocket. Color code of stick model: oxygen: red; nitrogen: blue; carbon: green; PLP: orange.

>>>Referee response

Thank you for rewriting this paragraph. Nevertheless, please add the corresponding citations for PyMOL and Spartan'16.

>>>Author response to Minor revisions.

...

>>>Referee response

The minor revisions have been addressed and adequately corrected. Thank you

Response to the referees' comments and revisions that have been made

We thank all of the reviewers for their comments. These have been very helpful in further improving the manuscript. We revised our manuscript in lights of all the comments as follows.

Reviewer #1

Original comments from Reviewer #1

The authors have addressed the comments made and have improved the manuscript significantly. The revised text does need some rewording, however. Suggestions are made in the new passages - see passages highlighted in turquoise color in the attached pdf.

Point-by-point response to the comments of Reviewer #1

We wish to thank the reviewer for the expert comment and for providing constructive suggestions. We have addressed all the points raised by the reviewer through new text.

- ✓ **Comment.** The authors have addressed the comments made and have improved the manuscript significantly. The revised text does need some rewording, however. Suggestions are made in the new passages - see passages highlighted in turquoise color in the attached pdf.

>>>Response to Comment.

We sincerely thank the reviewer for the constructive suggestions. Following the reviewer's suggestions, we have revised the corresponding sentences in the manuscript (highlighted in yellow). Additionally, we have further improved the manuscript by using the English editing service proposed by the editor.

[revMT_EnglishEditing.pdf, p7, line 4 – 9]

For example, in the case of the coupled assay for SHMT,^{11,15} SHMT first produces Gly and CH₂-THF from Ser and THF. Then, the coupled enzyme methylene-THF dehydrogenase uses CH₂-THF as a substrate to convert the coenzyme NADP⁺ to NADPH. By monitoring the conversion of NADP⁺ to NADPH by UV absorbance or fluorescence, SHMT activity can be indirectly detected.

[revMT_EnglishEditing.pdf, p4, line 17 – p5, line 11]

SHMT inhibitors have been developed with two primary biomedical goals in mind. The first goal is to develop new antimalarial drugs.^{3,9-15} Malaria is a life-threatening disease that spreads to people through infected anopheles mosquitoes. It has had a tremendous impact globally; 216 million people were infected in 2016, and 445,000 died.¹³ In addition, the resistance of malaria parasites against existing antimalarial drugs has become a serious problem. This drug resistance problem underscores the importance of efforts to develop new classes of inhibitors for the SHMT

enzyme associated with malarial parasites. The second goal is to develop novel anticancer drugs.^{1,2} There are two main isozymes of SHMT in mammalian cells; SHMT1 is present in the cytoplasm, and SHMT2 is present in mitochondria, as shown in **Fig. 1b**.⁴ In chemotherapy, the three enzymes of one-carbon metabolism, SHMT, dihydrofolate reductase (DHFR),¹⁶ and thymidylate synthase (TS),¹⁷⁻¹⁹ are all potential target enzymes as this pathway is central to pyrimidine biosynthesis and is therefore strongly related to cell proliferation (**Fig. 1c**). In fact, inhibitors targeting DHFR and TS, such as methotrexate and fluorouracil, respectively, have been used for a long time as anticancer agents. Among the three enzymes involved in one-carbon metabolism, to our knowledge, human SHMT (hSHMT) is the only enzyme for which an established chemotherapeutic agent has not yet been developed. Therefore, hSHMT has attracted attention as a potential target enzyme for inhibitor development.

[revMT_EnglishEditing.pdf, p6, Figure 1 caption]

Figure 1 | Biological role of SHMT. a) Serine–glycine interconversion catalyzed by SHMT. THF = tetrahydrofolate, CH₂-THF = N-5,N-10-methylenetetrahydrofolate. The red dot highlights the carbon that is transferred from Ser to THF. **b)** Schematic overview of hSHMT function. MTHFD = methylenetetrahydrofolate dehydrogenase-cyclohydrolase. CH₂-THF = N-5,N-10-methylenetetrahydrofolate. CH⁺-THF = 5,10-methenyltetrahydrofolate. CHO-THF = 10-formyltetrahydrofolate. NADP⁺ = Nicotinamide adenine dinucleotide phosphate. NADPH = NADP⁺ reduced form. **c)** SHMT, dihydrofolate reductase (DHFR), and thymidylate synthase (TS) in the folate cycle. THF = tetrahydrofolate, CH₂-THF = 5,10-methylenetetrahydrofolate, DHF = dihydrofolate, FdUMP = fluorodeoxyuridine-5'-monophosphate, dUMP = deoxyuridine monophosphate, dTMP = deoxythymidine monophosphate.

[revMT_EnglishEditing.pdf, p10, line 5–10]

NMR is also a promising detection method and has the advantage of having a high signal transparency even in opaque samples. Due to the transparency of NMR signals, NMR molecular probes are suitable for analysis under crude biological conditions. ^{19}F NMR, which is a nucleus with high sensitivity similar to that of ^1H , has been actively used in the design of NMR molecular probes for the analysis of biomolecules.^{32–40}

Reviewer #2

Original comments from Reviewer #2

This reviewer thanks the authors for their response. I am almost satisfied. I would still like the authors to evaluate the stability of their probe(s) against CYPs. In particular the authors can use newly purchased or prepared rat liver microsomes for these experiments.

Once this key experiment has been completed I would be supportive of publication.

Point-by-point response to the comments of Reviewer #2

We sincerely thank the reviewer for the positive comment for publication and constructive advice to improve our manuscript. We have conducted additional experiment to answer the question raised by the reviewer.

- ✓ **Comment.** This reviewer thanks the authors for their response. I am almost satisfied. I would still like the authors to evaluate the stability of their probe(s) against CYPs. In particular the authors can use newly purchased or prepared rat liver microsomes for these experiments.
Once this key experiment has been completed I would be supportive of publication.

>>>Response to Comment.

We sincerely thank the reviewer for the expert comment. Following the reviewer's comment, we have newly purchased rat liver microsomes (SIGMA, M9066-1VL: Microsomes from Liver, Pooled from Sprague-Dawley rat, male) and evaluated the stability of our probes against CYPs by HPLC. After 1 hour incubation of probe **1** and product DMANA with rat microsomes and NADPH at 37 °C, 98 ± 6 % of probe **1** and 79 ± 3 % of the product DMANA remained intact. These results give important suggestions about the stability of our probes in liver. This point has been added in the main text (p. 14, line 5–8) and new Supplementary Figure has been added in Supplementary information (**Supplementary Fig. S6c**).

[revMT_EnglishEditing.pdf, p14, line 5–9]

To evaluate the probe stability against cytochrome P450 (CYP450) enzymes, we incubated probe **1** and DMANA with NADPH-supplemented rat liver microsomes. Under our experimental conditions, majority of probe **1** and DMANA remained intact upon incubation with the microsomes (**Supplementary Fig. S6**).

a) Probe 1

b) DMANA

c)

New Supplementary Figure S6 | The stability of probe 1 and turned over product DMANA against reactive oxygen species, reactive species having thiols and amines, and rat microsomes. a) Fluorescence intensity change of probe 1 (1 μ M). b) Fluorescence intensity change of DMANA (5 μ M). Assay conditions: incubation time for 60 min at 37 $^{\circ}$ C. Excitation at 390 nm. 50 mM HEPES buffer (pH 7.5), 100 mM NaCl, 0.5

mM EDTA, 1 mM dithiothreitol (DTT), 1% DMSO. Fluorescence spectra were measured using a SHIMADZU RF-6000 fluorescence spectrometer (a) and using a JASCO FP-6500 fluorescence spectrometer (b). c) Metabolic stability assays of probe **1** (5 μ M) and turned over product DMANA (5 μ M). Assay conditions: incubation time for 60 min at 37 °C. 20 mM HEPES buffer (pH 7.5), rat liver microsomes (SIGMA, M9066-1VL: 200 μ g/mL final concentration with 5 μ M NADPH). Conversions were monitored using a SHIMADZU HPLC equipped with fluorescence detector RF-10A XL. Probe **1**: excitation at 340 nm, Detection at 435 nm. DMANA: excitation at 390 nm, Detection at 530 nm. Error bars represent s.d., n = 3.

Reviewer #3

Original comments from Reviewer #3

General response to the corrections

I would like to thank the authors for the careful and thorough corrections of the manuscript and for taking the addressed recommendations into consideration to improve the quality of the manuscript. The manuscript “Design Strategy for Serine Hydroxymethyltransferase Probes Based on Retro-Aldol-Type Reaction” from the Prof. Sando et al. fulfills the publication requirements for Nature Comm. with minor revisions.

Point to Point Response

>>>Author response.

We thank the reviewer for the constructive comment to improve this manuscript. Following the reviewer’s comment, we have revised the manuscript and added 25 references for covering related literatures appropriately. We hope that our revisions have adequately addressed the reviewer’s concerns.

>>>Referee response

Thank you for adding additional literature to the manuscript to cover the field.

>>>Author response.

We thank the reviewer for the constructive advice. Following the reviewer's comment, we have added new sentences about the advantage and precedents of ^{19}F NMR/MRI, with appropriate references (Revised manuscript, p.10, lines 5–10). We hope that our revisions have adequately addressed the reviewer’s concerns.

>>>Referee response

Thank you for the elaboration of ^{19}F NMR techniques.

>>>Author response.

Thank you for your valuable comment to improve this manuscript. Following the comments, we have added sentences to explain the importance of SHMT in the therapeutic area of Malaria and the previous efforts of anti-Malaria drug to the manuscript. [Revised

manuscript, p.4, lines 13–19] The development of SHMT inhibitors has been performed especially toward treatment of two types of diseases. The first one is antimalarial drug.8–15 Malaria is a life-threatening disease that spreads to people through infected anopheles mosquitoes. It has a tremendous impact globally, 216 million people are infected in 2016, and 445,000 people have died.¹³ In addition, the resistance of malaria parasites against existing antimalarial drugs has become a serious problem. Under such circumstances, researches and investigations for new inhibitors against malaria SHMT has been conducted.

>>>Referee response

I appreciate that malarial SHMT has been further introduced to the reader. But nevertheless, please rewrite this paragraph to underline the interesting and strong content with an adequate writing.

>>>Author response.

We thank the reviewer for the constructive comment. Following the reviewer's comment, we have revised the manuscript. We hope that our revisions have adequately addressed the reviewer's concerns. If the reviewer needs more rewriting, we are ready to rewrite again.

>>>Referee response

Please rewrite the introduction. It is hard to read and not harmonized yet. No changes has been conducted compared to previous version.

>>>Author response.

We agree with the reviewer's comment. The product aldehyde might react with biological relevant nucleophiles through forming Schiff's base. Following the reviewer's comment, first we have checked the stability of fluorescence product aldehyde (DMANA) against biological reactants, e.g. glutathione, L-cysteine, DL-homocysteine, H₂O₂, NaOCl, KO₂, TBHP, and NOC7 (Supplementary Fig. S6). Under all conditions, no significant changes in fluorescence were observed. Furthermore, we have evaluated the toxicity of DMANA (Supplementary Fig. S7). Significant toxicity was not observed from the cytotoxicity test. These results indicate that the significant side-effects and toxicity are not problematic under the concentration and time range of fluorescence experiments. We have added the discussion about the stability and toxicity of DMANA in the main text and Supplementary Information (p. 14, lines 1–8, Supplementary Fig. S6, Supplementary Fig. S7). However, under the concentration range of ¹⁹F NMR/MRI, ¹⁹F product aldehyde might show the

side-effects and toxicity, because the required concentration is high for detection of ^{19}F NMR/MRI signal. For practical in vivo MRI, it might be necessary to improve the sensitivity to lower the required concentration. These points have also been discussed in the main text (p. 24, lines 15–19).

Supplementary Figure S6 | The stability of a) probe 1 and b) turned over product DMANA against reactive oxygen species and reactive species having thiols and amines. a) Fluorescence intensity change of probe 1 (1 μM). b) Fluorescence intensity change of DMANA (5 μM). Assay conditions: Incubation time for 60 min at 37 °C. Excitation at 390 nm. 50 mM HEPES buffer (pH 7.5), 100 mM NaCl, 0.5 mM EDTA, 1 mM dithiothreitol (DTT), 1% DMSO. Fluorescence spectra were measured using a SHIMADZU RF-6000 fluorescence spectrometer (a) and using a JASCO FP-6500 fluorescence spectrometer (b). Error bars represent s.d., $n = 3$.

Supplementary Figure S7 | Cytotoxicity evaluation of turned over fluorescence product DMANA. HeLa cells were plated in a 6-well plate at a density of 50,000 cells/mL in DMEM media. After incubation for 24 h, each samples were supplemented with either 1 μM or 5 μM DMANA, containing 1% DMSO. Equivalent samples were supplemented with 1% DMSO as a vehicle control. At 6 and 24 hours, cells were dissociated from wells by trypsin, a 10 μL sample was removed from each of the samples and mixed 1:1 with a 0.4% wt/volume trypan blue solution in PBS. Samples were incubated for 1 minute at room temperature before being loaded onto a hemocytometer where live and dead cells were counted. Each sample was made in triplicate for each time point. Error bars represent s.d., $n = 3$.

>>>Referee response

I appreciate the effort to answer my question about stability and toxicology. The question is very well addressed, which further improves the quality of the manuscript.

>>>Author response.

Human SHMT shares 91% sequence identity with mouse SHMT and 42% sequence identity with *P. vivax* SHMT (PvSHMT) (Chaiyen et al., FEBS J., 276, 4023–4036 (2009)). Because human and mouse have a high homology, mouse SHMT is considered to be

appropriate for this modeling study. We have added this point about homology in the main text (P8, lines 5–8).

>>>Referee response

Thank you for further elucidation of the homology of the species. Nevertheless, a short comment about the homology of the active site would be appropriate.

>>>Author response.

Basically, we think that it is difficult to conduct in vivo experiments by using the current probes and machine setup. However, regarding fluorescence probe 1, depending on the two-photon efficiency of the turned-over product, it might be possible to detect the product near the surface of the mouse/rat by using a two-photon excitation microscope. Regarding practical in vivo ¹⁹F MRI using probe 2, there is a problem of sensitivity, so it would be necessary to enhance the sensitivity, for example, by increasing the number of ¹⁹F atoms on molecular probes or by improving NMR/MRI instruments. In order to clarify this point, we have added this discussion about current limitation and future direction in an application for ¹⁹F MRI in main text. (P24, lines 15–19)

>>>Referee response

As this point was not only mentioned by me, it was very necessary to clarify the concerns. I appreciate the additional comment.

>>>Author response.

We thank the reviewer for the comment. Following the reviewer's comment, we have added an explanation of manual modeling process as follows (caption in Supplementary Fig. S3). We hope that our revisions have adequately addressed the reviewer's concerns. [Revised Supplementary Information, p.S5] Supplementary Figure S3 | Illustration of probe 1 manually fitted in active site of mSHMT1 (PDB ID: 1EJI). First, the calculation of global minimum energy conformation of L-erythro probe 1 was performed by Spartan'16 (Wavefunction, Inc.) software. Then, the co-crystal structure of PLP-Gly-5-CHO-THF-mSHMT1 (PDB ID: 1EJI) and probe 1 are displayed using the PyMOL software, and 5-CHO-THF is deleted. Finally, L-erythro probe 1 was manually placed to the active site of mSHMT1 with PLP-Gly as amino groups and carboxylic groups of Gly and probe 1 were each superimposed. This supports the hypothesis that L-erythro form can be accommodated in the substrate pocket. Color code of stick model:

oxygen: red; nitrogen: blue; carbon: green; PLP: orange.

>>>Referee response

Thank you for rewriting this paragraph. Nevertheless, please add the corresponding citations for PyMOL and Spartan'16.

>>>Author response to Minor revisions.

...

>>>Referee response

The minor revisions have been addressed and adequately corrected. Thank you

Point-by-point response to the comments of Reviewer #3

We sincerely thank the reviewer for the constructive, valuable, expert comments to improve the manuscript. We have addressed all the points raised by the reviewer through new text.

- ✓ **Comment.** I appreciate that malarial SHMT has been further introduced to the reader. But nevertheless, please rewrite this paragraph to underline the interesting and strong content with an adequate writing.

>>>Response to Comment.

We sincerely thank the reviewer for the constructive comments. According to reviewer 1 and 3's comments, we have rewritten the paragraph concerning attractive target SHMT in antimalarial and anticancer drug development (revMT_EnglishEditing.pdf, p4 line 17 – p5 line 11). In addition, the English writing of this revised manuscript was checked and improved by the English editing service proposed by the editor. We hope that our revisions have adequately addressed the reviewer's concerns.

- ✓ **Comment.** Please rewrite the introduction. It is hard to read and not harmonized yet. No changes has been conducted compared to previous version.

>>>Response to Comment.

According to reviewer's comment, we have rewritten the introduction. To improve the readability, we have added a new 1st paragraph describing the importance of SHMT in one-carbon metabolism. In addition, we have revised the introduction part carefully to harmonize.

In addition, the English writing of this revised manuscript was checked and improved by the English editing service proposed by the editor. We hope that our revisions have adequately addressed the reviewer's concerns.

- ✓ **Comment.** Thank you for further elucidation of the homology of the species. Nevertheless, a short comment about the homology of the active site would be appropriate.

>>>Response to Comment.

According to reviewer's comment, we have added the sentence about the homology of the active site.

[revMT_EnglishEditing.pdf, p8, line 7–9]

The active site residues of the human and mouse SHMTs are nearly identical, and the active site residues of the human and malaria SHMTs are about 80% similar.^{23,24}

- ✓ **Comment.** Thank you for rewriting this paragraph. Nevertheless, please add the corresponding citations for PyMOL and Spartan'16.

>>>Response to Comment.

We thank the reviewer for this advice. According to reviewer's comment, we have added the corresponding citations.

[Revised Supplementary Information, pS32, Reference section]

(1) Hehre, W., Ohlinger, S. The Spartan'16 Tutorial and User's Guide, Wavefunction, Inc., Irvine, CA, USA (2016).

(2) DeLano, W. L. The Pymol User's Manual. Delano Scientific, San Carlos, CA, USA (2002).